# The p97/VCP adaptor UBXD1 drives AAA+ remodeling and ring opening through multi-domain tethered interactions

Julian R. Braxton [1,2,5], Chad R. Altobelli[1,3], Maxwell R. Tucker [2,4], Eric Tse [2], Aye C. Thwin[2], Michelle R. Arkin [3] ✉ & Daniel R. Southworth [2] ✉

p97, also known as valosin-containing protein, is an essential cytosolic AAA+ (ATPases associated with diverse cellular activities) hexamer that unfolds substrate polypeptides to support protein homeostasis and macromolecular disassembly. Distinct sets of p97 adaptors guide cellular functions but their roles in direct control of the hexamer are unclear. The UBXD1 adaptor localizes with p97 in critical mitochondria and lysosome clearance pathways and contains multiple p97-interacting domains. Here we identify UBXD1 as a potent p97 ATPase inhibitor and report structures of intact human p97–UBXD1 complexes that reveal extensive UBXD1 contacts across p97 and an asymmetric remodeling of the hexamer. Conserved VIM, UBX and PUB domains tether adjacent protomers while a connecting strand forms an N-terminal domain lariat with a helix wedged at the interprotomer interface. An additional VIM-connecting helix binds along the second (D2) AAA+ domain. Together, these contacts split the hexamer into a ring-open conformation. Structures, mutagenesis and comparisons to other adaptors further reveal how adaptors containing conserved p97-remodeling motifs regulate p97 ATPase activity and structure.

p97/VCP is a AAA+ unfoldase involved in many cellular processes, including membrane fusion, autophagic clearance of damaged organelles and protein homeostasis[1,2]. Missense mutations in p97 cause multisystem proteinopathy (also called IBMPFD), amyotrophic lateral sclerosis and vacuolar tauopathy, and disrupt many p97-dependent functions[3–5]. The central mechanism across these diverse activities is the extraction of proteins from macromolecular complexes and membranes through hydrolysis-driven substrate translocation by p97 (refs. 6,7). In achieving these functions, p97 is regulated by more than 30 adaptor proteins (also known as cofactors) that facilitate substrate delivery, control subcellular localization and couple additional substrate-processing functions[8,9].

p97 forms homohexamers that enclose a central channel through which substrates are translocated and unfolded; each protomer consists of two AAA+ ATPase domains (D1 and D2), an N-terminal domain (NTD) and an unstructured C-terminal tail[10]. Substrate-bound p97 structures[6,7,11] revealed that the hexamer adopts a right-handed spiral with conserved pore loops in D1 and D2 that extend into the channel and contact the substrate polypeptide in a manner similar to many AAA+ translocases[12–14]. Notably, substrate recognition and engagement are dependent on p97–adaptor coordination[8], yet with the exceptions of UFD1–NPL4 (refs. 7,15,16) and p37 (refs. 17–19), little is known about the mechanisms by which adaptors control p97 activity.

¹Graduate Program in Chemistry and Chemical Biology, University of California San Francisco, San Francisco, CA, USA. ²Department of Biochemistry and Biophysics and Institute for Neurodegenerative Diseases, University of California San Francisco, San Francisco, CA, USA. ³Department of Pharmaceutical Chemistry and Small Molecule Discovery Center, University of California San Francisco, San Francisco, CA, USA. ⁴Graduate Program in Biophysics, University of California San Francisco, San Francisco, CA, USA. ⁵Present address: Division of Biology and Biological Engineering, California Institute of Technology, Pasadena, CA, USA. ✉e-mail: michelle.arkin@ucsf.edu; daniel.southworth@ucsf.edu

UBXD1 is a p97 adaptor associated with autophagy and endo-somal trafficking[20–24]. p97 mutations that are implicated in multisystem proteinopathy impair UBXD1 binding and associated cellular functions[22,25,26] but its role in regulating p97 in these pathways is unknown. UBXD1 contains three conserved p97 interaction motifs: a VCP-interacting motif (VIM) helix and ubiquitin regulatory X (UBX) domain, which canonically interact with the NTD, and a PUB domain (peptide:N-glycanase and UBA-containing or UBX-containing proteins) that interacts with the C-terminal tail[24,27–30]. The UBX domain is reported to not interact with p97 owing to the mutation of key residues[20,27,31]. Rather, this domain has been reported to bind ubiquitin and may thus function in substrate binding[32]. UBXD1 is the only known adaptor with both p97 N- and C-terminal-interacting domains; this feature, in addition to several other potential p97 binding elements, raises key questions about the arrangement of UBXD1 on p97 hexamers and its effect on p97 structure and function.

To address these questions, we determined cryo-electron microscopy (cryo-EM) structures of p97 in complex with UBXD1. In the structures, extensive UBXD1 interactions are identified across three p97 protomers that together remodel the p97 hexamer into distinct closed-ring and open-ring states in which the remodeled protomers are tethered by UBXD1 contacts. A lariat structure wraps around an NTD and functions as a wedge that displaces D1 interprotomer contacts, while a short helix adjacent to the VIM alters the D2 conformation. UBXD1 interactions coincide with potent p97 ATPase inhibition. Together with mutagenesis and additional structural characterization, this work reveals distinct roles for UBXD1 domains and identifies how adaptor interactions coordinate to remodel p97 structure and control function.

## Results

### Structures of p97–UBXD1 reveal extensive hexamer remodeling

With three canonical p97 binding domains in UBXD1 (VIM, UBX and PUB), we postulated that substantial interactions with p97 may define its function. UBXD1 also contains transiently structured helices, H1/H2 (residues 1–25) and H4 (residues 75–93), that weakly interact with p97 (refs. 24,25) and a helical lariat with homology to the adaptor alveolar soft part sarcoma locus (ASPL, residues 278–334) (refs. 32,33). Analysis of the UBXD1 AlphaFold model[34,35] revealed structures for the canonical domains, H1/H2, H4 and the helical lariat; the remaining sequence appears unstructured (Fig. 1a,b).

We first determined the effect of UBXD1 on p97 ATPase activity, finding potent inhibition of p97 with a half-maximal inhibitory concentration ($IC_{50}$) of 25 nM (Fig. 1c). We next analyzed UBXD1 binding to p97 by size exclusion chromatography (SEC) and SDS–PAGE analysis. Following incubation with either ADP or the slowly hydrolyzable ATP analog ATPγS, UBXD1 co-elutes with p97, indicating a stable interaction (Extended Data Fig. 1a,b). However, based on the reduced band intensity of UBXD1 compared to p97 in both conditions, UBXD1 appears to be sub-stoichiometric with respect to p97. Based on these results and the reported structural stability of the NTDs in the ADP state[36], p97–UBXD1 cryo-EM samples were first prepared with ADP (Extended Data Fig. 1c). Reference-free 2D class averages show well-defined top and side views of the p97 hexamer with the NTDs in the down conformation, co-planar with the D1 ring, but with no density attributable to UBXD1 (Fig. 1d and Extended Data Fig. 1d). Next, 3D classification was performed (Extended Data Fig. 1e), resulting in the identification of two states with prominent UBXD1 density (p97–UBXD1[closed] and p97–UBXD1[open]) that were subsequently refined (Fig. 1e,f and Table 1). Individual p97 protomers are denoted P1–P6, counterclockwise, based on the asymmetry of p97–UBXD1[open]. High-resolution features in both states enabled the building of atomic models, with all UBXD1 domains identified except H1/H2 and H4 (Fig. 1e,f, Extended Data Fig. 2a–d, k and Supplementary Video 1).

In both structures, density for ADP is present in all D1 and D2 nucleotide pockets and coordinated by conserved AAA+ residues (Extended Data Fig. 1f–h). In p97–UBXD1[closed], D1 and D2 are in a planar conformation and one molecule of UBXD1 is identified, binding across protomers P1 and P6. In p97–UBXD1[open], a single UBXD1 is similarly bound but the interface between P1 and P6 is separated by ~8 Å relative to p97–UBXD1[closed], creating an open-ring arrangement of the p97 hexamer. For both structures, strong density corresponding to the VIM and UBX domains is observed, as well as for the helical lariat that encircles the NTD of the P6 protomer (Fig. 1e,f). At a lower density threshold, the PUB domain becomes apparent and is positioned below the UBX, adjacent to the p97 D2 ring, but appears to make no substantial contact with the hexamer, probably resulting in substantial flexibility and explaining the lower resolution (Fig. 1g). Weak density resembling the VIM is also present in the NTDs of P2–P5 in both states, indicating partial binding to these protomers, which is probably a consequence of protein concentrations used for cryo-EM (Extended Data Fig. 1i). Overall, these structures identify that a singly bound UBXD1 drives ring opening at a protomer interface that is tethered by multiple UBXD1 interactions involving the VIM, UBX and lariat motif. We identify that the UBXD1 interaction buries a surprising ~3,200 Å² of surface area, the largest of any p97–adaptor interaction that is currently structurally characterized.

During 3D classification, additional structures with different UBXD1 configurations were identified (Extended Data Figs. 1e, 2 and 3 and Table 1). A prevalent class, termed p97–UBXD1[VIM], closely resembles the previously determined structure of the p97 hexamer bound to ADP (p97[ADP], Cα root mean square deviation (r.m.s.d.) of 1.0 Å)[36] but features additional helical density in the NTD cleft of all protomers that corresponds to the UBXD1 VIM (Extended Data Fig. 3a). Two additional states were identified, termed p97–UBXD1[meta] and p97–UBXD1[para], in which densities for the VIM, PUB, lariat and UBX are also observed at other positions (across P2–P3 or P3–P4, respectively) in the p97 hexamer in conformations similar to p97–UBXD1[closed], indicating that other configurations can occur (Extended Data Fig. 3b,c). However, the p97 open ring is observed only in the singly bound UBXD1 configuration, indicating that this conformation is driven by binding of one UBXD1. Considering the singly bound closed and open states exhibit substantial p97 conformational changes and extensive interactions by UBXD1, these states are largely the subsequent focus of this study.

### UBXD1 promotes nucleotide-dependent p97 conformational changes

To identify the effects of UBXD1 binding, p97–UBXD1[closed] and p97–UBXD1[open] were aligned to p97[ADP36]. The r.m.s.d. values reveal extensive changes across D1 and D2 for the seam protomers (P1 and P6) in both UBXD1-bound states, although the magnitudes are greater in the open state (Fig. 2a,b, Extended Data Fig. 4a,b and Supplementary Video 2). The open state features a modest right-handed spiral with an overall elevation change of 7 Å (Extended Data Fig. 4c), largely owing to a 9° downward rotation of P1 (Fig. 2b). In addition to movements between protomers, there is a notable rotation between the small and large subdomains of D2 in protomer P1 in both states (Extended Data Fig. 4d). This rotation is particularly evident in the open state, in which the small subdomain is rotated upward by 10° relative to p97[ADP] (Extended Data Fig. 4d). This rotation, and the separation of P1 and P6, causes a helix (α5′) (refs. 10,37), which is normally positioned on top of α12′ of the D2 domain of the counterclockwise protomer, to disappear from the density map, probably due to increased flexibility (Extended Data Fig. 4e and Fig. 2a,b).

The closed and open states each show distinct P1–P6 interfaces (Extended Data Fig. 4f). In the closed state, the D1 domains of P1 and P6 are separated by 3 Å relative to p97[ADP], while there is negligible D2 separation. In the open state, the D1 and D2 domains are separated by 8 Å and 11 Å, respectively. These changes remodel the nucleotide-binding

pockets of P1 by displacing the arginine fingers from P6 (Extended Data Fig. 4d), probably precluding ATP hydrolysis in P1. To further investigate the conformational changes associated with UBXD1 binding, 3D variability analysis (Methods) was performed jointly on particles from the closed and open states (Supplementary Video 3). This reveals the transition from a planar UBXD1-bound hexamer to a partial spiral, indicating that the closed and open states may be in equilibrium and on path to substrate-bound conformations[6,7].

We next sought to determine structures of p97–UBXD1 with ATPγS in order to identify nucleotide-dependent differences in UBXD1 interactions. Cryo-EM analysis of p97–UBXD1 incubated with ATPγS revealed three predominant classes: a UBXD1-bound hexamer similar to the ADP-bound closed state, a symmetric hexamer lacking UBXD1 density with NTDs in the 'up', ATP conformation and a symmetric hexamer similar to p97–UBXD1[VIM] (Extended Data Fig. 5a,b). An open state was not present in the dataset, indicating that this conformation may require a post-hydrolysis ADP state of p97. Despite the unambiguous presence of ATPγS in all nucleotide-binding pockets (Extended Data Fig. 5c), the NTDs adopt the 'down' conformation in maps containing UBXD1 density (the closed and VIM-only states), indicating that UBXD1 interactions favor the NTD 'down' state and appear to override the 'up' conformation observed in ATP-bound structures[36].

We postulate that the symmetric VIM-only class in the ADP and ATPγS datasets (Fig. 2c and Extended Data Figs. 1e and 5a) may be a consequence of our incubation conditions involving a high, 2:1 concentration of UBXD1 relative to the p97 monomer. We predict that this results in saturated occupancy of the NTDs by the VIM helix and displacement of other interactions that define the asymmetric closed and open states. Therefore, we sought to examine UBXD1-promoted rearrangements in conditions more similar to those in vivo, in which p97 is present in excess of UBXD1 (ref. 38). To that end, we incubated p97 with a substoichiometric amount of UBXD (1 UBXD1 per p97 hexamer) in the presence of ADP and determined the resulting distribution of complexes by cryo-EM (Extended Data Fig. 5d–f and Methods). This analysis revealed that ~75% of the particles have no UBXD1 density, as expected given the reduced UBXD1 concentration (Fig. 2c). Strikingly, the remaining 25% are in the open state with a singly bound UBXD1, indicating that under substoichiometric conditions, UBXD1 binds predominantly as one per hexamer in the ring-open conformation. These data further support the functional significance of the open conformation, wherein binding by one UBXD1 through multiple interactions breaks the p97 interprotomer interface and opens the hexamer ring. Taken together, our findings indicate that UBXD1 may function

during the p97 catalytic cycle by promoting ring opening specifically in a post-hydrolysis state (Fig. 2d).

## Canonical UBXD1 interactions across three p97 protomers

The p97–UBXD1[closed] and p97–UBXD1[open] structures reveal how the conserved VIM, UBX and PUB domains of UBXD1 interact simultaneously across the p97 hexamer. In contrast to previous studies[20,27,31], both VIM and UBX are identified to interact with p97, binding adjacent NTDs (Fig. 3). The 18-residue VIM helix is positioned in the NTD cleft of P1, similar to structures of isolated domains, and comprises the only major contact with this protomer (Fig. 3a,b and Extended Data Fig. 6a,b). A conserved arginine residue (R62), required for p97 binding[29], projects into the NTD, potentially forming a salt bridge with D35 of the NTD and

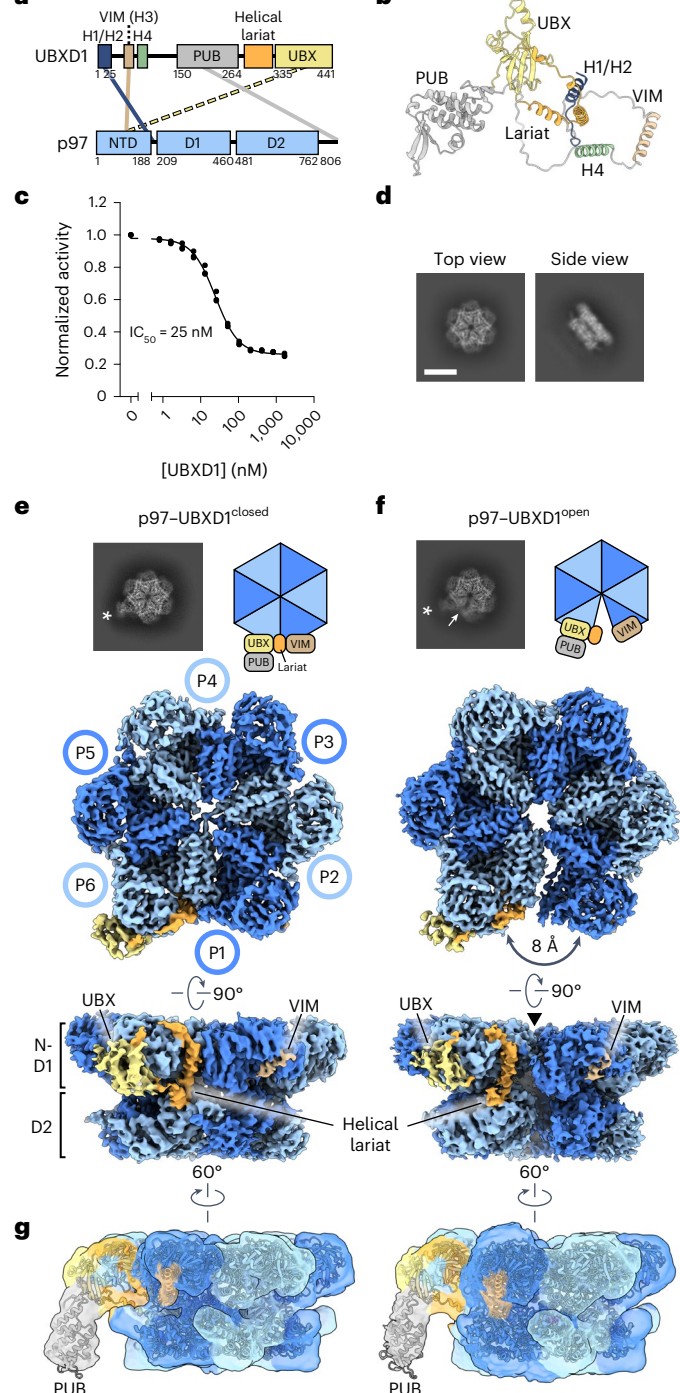

**Fig. 1 | Cryo-EM structures of p97–UBXD1 closed and open states. a**, Domain schematics of UBXD1 and p97 (not to scale) showing reported interactions (solid lines) between conserved domains[24,27–29] and the UBX–NTD interaction previously reported to not occur for UBXD1 (dashed line)[20,27,31]. **b**, AlphaFold model of UBXD1 showing structured regions (H1/H2, VIM, H4, PUB, helical lariat and UBX) colored as in **a**. **c**, Steady-state ATPase activity (*y* axis, normalized to activity at 0 nM UBXD1) of p97 at increasing concentrations of UBXD1 (*x* axis), resulting in a calculated $IC_{50}$ of 25 nM. Data are from *n* = 3 independent experiments, each with three technical replicates. Data are presented as mean values from each independent experiment. **d**, Representative 2D class averages following the initial classification of the full p97–UBXD1 dataset, showing the p97 hexamer and no additional density for UBXD1. Scale bar, 100 Å. **e**,**f**, Final cryo-EM reconstructions of p97–UBXD1[closed] (**e**) and p97–UBXD1[open] (**f**) states with top-view 2D projections showing UBX–PUB density (*) and open p97 ring (arrow) compared to cartoon depictions of the corresponding complexes (top row); cryo-EM density maps (p97–UBXD1[open] is a composite map; see Methods), colored to show the p97 hexamer (light and dark blue, with protomers labeled P1–P6) and UBXD1 density for the VIM (brown), UBX (yellow) and lariat (orange) domains (bottom row). The 8 Å separation between protomers P1 and P6 is indicated for p97–UBXD1[open]. **g**, Low-pass filtered maps and fitted models of p97–UBXD1[closed] (left) and p97–UBXD1[open] (right) exhibiting low-resolution density for the PUB domain (gray).

**Table 1 | Cryo-EM data collection, refinement and validation statistics**

| | p97–UBXD1closed (EMD-28982, PDB 8FCL) | p97–UBXD1open (EMD-28983, PDB 8FCM) | p97–UBXD1VIM (EMD-28987, PDB 8FCN) | p97–UBXD1meta (EMD-28988, PDB 8FCO) | p97–UBXD1para (EMD-28989, PDB 8FCP) | p97–UBXD1-PUBin (EMD-28990, PDB 8FCQ) | p97–UBXD1H4 (EMD-28991, PDB 8FCR) | p97–UBXD1LX (EMD-28992, PDB 8FCT) |
|---|---|---|---|---|---|---|---|---|
| **Data collection and processing** | | | | | | | | |
| Microscope and camera | Titan Krios, K3 | Titan Krios, K3 | Titan Krios, K3 | Titan Krios, K3 | Titan Krios, K3 | Titan Krios, K3 | Titan Krios, K3 | Titan Krios, K3 |
| Magnification (×) | 59,952 | 59,952 | 59,952 | 59,952 | 59,952 | 59,952 | 59,952 | 59,952 |
| Voltage (kV) | 300 | 300 | 300 | 300 | 300 | 300 | 300 | 300 |
| Data acquisition software | SerialEM | SerialEM | SerialEM | SerialEM | SerialEM | SerialEM | SerialEM | SerialEM |
| Exposure navigation | Image shift | Image shift | Image shift | Image shift | Image shift | Image shift | Image shift | Image shift |
| Electron exposure (e⁻/Å²) | 43 | 43 | 43 | 43 | 43 | 43 | 43 | 43 |
| Defocus range (μm) | −0.5 to −2.0 | −0.5 to −2.0 | −0.5 to −2.0 | −0.5 to −2.0 | −0.5 to −2.0 | −0.5 to −2.0 | −0.5 to −2.0 | −0.5 to −2.0 |
| Pixel size (Å) | 0.834 | 0.834 | 0.834 | 0.834 | 0.834 | 0.834 | 0.834 | 0.834 |
| Symmetry imposed | C1 | C1 | C6 | C1 | C2 | C1 | C1 | C1 |
| Initial particle images (no.) | 5,498,937 | 5,498,937 | 5,498,937 | 5,498,937 | 5,498,937 | 5,498,937 | 5,498,937 | 2,250,090 |
| Final particle images (no.) | 82,334 | 563,468 | 100,000 | 80,700 | 45,628 | 59,126 | 24,086 | 106,993 |
| Map resolution (Å) | 3.5 | 3.3 (consensus) | 3.0 | 3.3 | 3.5 | 3.9 | 4.1 | 3.4 |
| FSC threshold | 0.143 | 0.143 | 0.143 | 0.143 | 0.143 | 0.143 | 0.143 | 0.143 |
| Map resolution range (Å) | 2–10 | 2–10 (consensus) | 2–10 | 2–10 | 2–10 | 3–12 | 3–12 | 2.5–10 |
| **Refinement** | | | | | | | | |
| Initial model used (PDB, AlphaFold code) | PDB 5FTK, AF-Q9BZV1-F1 | PDB 5FTK, AF-Q9BZV1-F1 | PDB 5FTK, AF-Q9BZV1-F1 | PDB 5FTK, AF-Q9BZV1-F1 | PDB 5FTK, AF-Q9BZV1-F1 | PDB 5FTK, AF-Q9BZV1-F1 | PDB 5FTK, AF-Q9BZV1-F1 | PDB 5FTK, AF-Q9BZV1-F1 |
| Model resolution (Å) | 4.3 | 3.8 | 3.4 | 4.2 | 4.1 | 6.2 | 6.2 | 4.0 |
| FSC threshold | 0.5 | 0.5 | 0.5 | 0.5 | 0.5 | 0.5 | 0.5 | 0.5 |
| Map sharpening B factor (Å²) | −93.5 | −86.1 (consensus) | −113.6 | −96.8 | −108.4 | −90.4 | −133.3 | −107.0 |
| Model composition | | | | | | | | |
| Non-hydrogen atoms | 36,904 | 36,829 | 35,034 | 39,569 | 39,494 | 36,904 | 36,931 | 34,329 |
| Protein residues | 4,667 | 4,658 | 4,446 | 4,999 | 4,990 | 4,667 | 4,679 | 4,356 |
| Ligands | 12 | 12 | 12 | 12 | 12 | 12 | 12 | 12 |
| B factors (Å²) | | | | | | | | |
| Protein | 30.09 | 30.08 | 27.06 | 33.80 | 33.80 | 30.09 | 30.19 | 26.10 |
| Ligand | 13.34 | 13.34 | 13.34 | 13.34 | 13.34 | 13.34 | 13.34 | 13.34 |
| r.m.s. deviations | | | | | | | | |
| Bond lengths (Å) | 0.011 | 0.011 | 0.011 | 0.011 | 0.011 | 0.011 | 0.011 | 0.011 |
| Bond angles (°) | 1.087 | 1.064 | 1.040 | 1.091 | 1.092 | 1.063 | 1.067 | 1.040 |
| **Validation** | | | | | | | | |
| MolProbity score | 0.72 | 0.83 | 0.71 | 0.82 | 0.81 | 0.83 | 0.87 | 0.71 |
| Clashscore | 0.67 | 1.16 | 0.65 | 1.13 | 1.07 | 1.15 | 1.40 | 0.62 |
| Poor rotamers (%) | 0.10 | 0.00 | 0.11 | 0.00 | 0.00 | 0.03 | 0.00 | 0.05 |
| Ramachandran plot | | | | | | | | |
| Favored (%) | 98.17 | 98.08 | 98.66 | 98.27 | 98.12 | 98.23 | 98.19 | 98.52 |
| Allowed (%) | 1.79 | 1.77 | 1.20 | 1.65 | 1.76 | 1.70 | 1.70 | 1.34 |
| Disallowed (%) | 0.04 | 0.15 | 0.14 | 0.08 | 0.12 | 0.06 | 0.11 | 0.14 |

a hydrogen bond with the backbone carbonyl of A142 (ref. 29). The VIM appears anchored at its N terminus by a salt bridge between E51 of UBXD1 and K109 of the NTD and by hydrogen bonding between the backbone carbonyl of E51 and Y143 of the NTD. Additional hydrophobic contacts could further stabilize this interaction (Fig. 3b, right).

The UBX domain is bound to the NTD of P6 in a manner similar to that of other adaptors despite mutation of the canonical phenylalanine–proline–arginine motif[8] located on the S3/S4 loop to serine–glycine–glycine (Fig. 3c,d and Extended Data Fig. 6c,d). However, another arginine residue that is important for interaction with p97, R342, is conserved in UBXD1 and probably forms a hydrogen bond with the backbone carbonyl of P106. The UBX domain features an additional β-strand (Uβ0) that is positioned proximal to the N-terminal lobe of the p97 NTD (Fig. 3e). Uβ0 directly connects the UBX, PUB and helical lariat, indicating that these three domains may function together. Additionally, a C-terminal extension consisting of two α-helices (Uα2 and Uα3) connected by unstructured linkers is positioned on the apical surface of the canonical UBX and wraps over the core β-sheet.

PUB domains bind the HbYX (hydrophobic, tyrosine, any amino acid) motif located at the end of the p97 C-terminal tail[28]. Density for this domain is more poorly resolved in both the closed and open structures, which prompted us to perform a focused classification of this region (Extended Data Fig. 1e). Two classes show improved definition for the PUB, enabling the AlphaFold model for this region to be fit unambiguously into the density (Extended Data Fig. 6e,f). In class 1 (p97–UBXD1-PUB_out), the PUB domain is positioned similarly to that in the closed model, and in class 2 (p97–UBXD1-PUB_in), the PUB domain is rotated 46° about a linker connecting the PUB and UBX domains and points towards P6 (Fig. 3f, Extended Data Figs. 2 and 6f,g and Table 1). In both classes, connecting density is observed between the PUB and the bottom surface of P6, indicative of binding to the P5 C-terminal tail (Fig. 3g,h and Extended Data Fig. 6h–k). However, stronger tail density in p97–UBXD1-PUB_in may indicate that binding of the HbYX motif is associated with inward rotation of the PUB domain. In summary, we identify that a single molecule of UBXD1 surprisingly interacts across three p97 protomers (P1, P5 and P6) through interactions by the VIM, UBX and PUB domains (Fig. 3i).

## The UBXD1 helical lariat and H4 make distinct p97 interactions

The UBXD1 helical lariat is among the most striking UBXD1 binding elements owing to its intimate interaction with all three domains of the P6 protomer (Fig. 4a–e and Extended Data Fig. 7a). This domain is composed of four helices (Lα1–Lα4) inserted between Uβ0 and Uβ1 of the UBX domain that completely encircle the P6 NTD (Fig. 4a). Lα1 and Lα2 are positioned along the top of the P6 NTD; Lα1 is poorly resolved, and Lα2 projects two phenylalanine residues into the NTD (Fig. 4c). Lα3 is situated at the D1 interface of P6 and P1 and makes electrostatic contacts with residues in both the N-terminal and D1 domains of P6; in the closed state hydrophobic contacts are also made with the D1 domain of P1 (Fig. 4b,d). Lα4 contacts the P6 D2 domain and connects back to the UBX domain, completing the lariat. A short loop connects Lα3 and Lα4, further anchoring the lariat into this D2 domain (Fig. 4b,e). Additionally, the Lα3–Lα4 arrangement appears stabilized by a tripartite electrostatic network (Fig. 4e). Considering Lα3 displaces typical D1–D1 contacts between P1 and P6, we postulate that this helix probably contributes substantially to the D1 conformational changes and ring opening identified in the closed and open states of p97. Interestingly, the binding site of the Lα3–Lα4 linker overlaps with that of the p97 allosteric inhibitor UPCDC30245 (ref. 36), suggesting that occupancy of this site is a productive means to alter p97 function (Extended Data Fig. 7b).

Further classification of p97–UBXD1^closed was performed to potentially resolve additional regions that were reported to interact with p97 (refs. 24,25) (Extended Data Fig. 1e). This analysis revealed an additional state similar to p97–UBXD1^closed but with low-resolution density on top of the D2 domain of the P1 protomer (Fig. 4f, Extended Data Figs. 2 and 7c

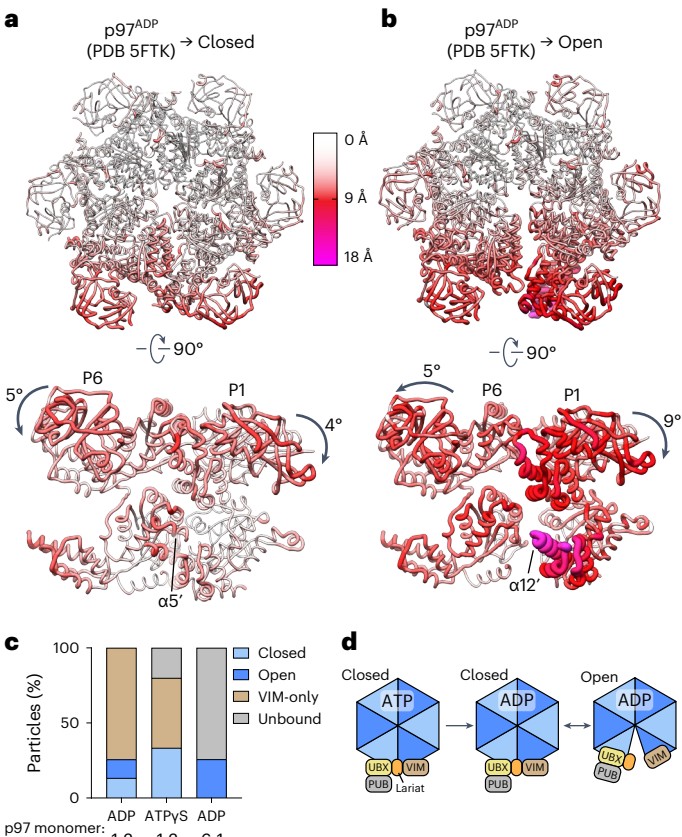

**Fig. 2 | UBXD1-mediated p97 hexamer remodeling. a,b,** The p97 hexamer and rotated side view of the seam protomers P1 and P6 for p97–UBXD1^closed (**a**) and p97–UBXD1^open (**b**) structures, colored according to Cα r.m.s.d. values relative to the p97^ADP symmetric state (PDB 5FTK, aligned to P3 and P4). The largest changes (>15 Å, magenta with wider tubes) are identified for α12′ of P1 (labeled) in the open state, intermediate changes (~10 Å, red) for P1 and P6 with rotations of the NTDs relative to p97^ADP shown and small to no changes for the remaining regions (<5 Å, white); α5′ of P6, which disappears in the open state, is labeled in **a**. **c,** Distribution of states for the ADP and ATPγS superstoichiometric datasets (2:1 UBXD1:p97 monomer) and the ADP substoichiometric dataset (1:6 UBXD1:p97 monomer). For each dataset, all particles after 2D classification were subjected to 3D classification using the same references. Classes corresponding to junk particles were excluded and the proportions of the remaining classes were plotted. **d,** Schematic of p97 remodeling in various nucleotide-bound and UBXD1-bound states.

and Table 1). This region probably corresponds to H4 because of its proximity to the VIM, which is predicted to be connected to H4 by a seven-residue linker (Fig. 1b). In this structure (p97–UBXD1^H4), H4 binding is associated with an upward rotation of the D2 small subdomain by ~17° relative to the closed state (Fig. 4g and Supplementary Video 4). This rotation results in the displacement of a short helix in P6 (α5′), weakening D2–D2 contacts between the seam protomers. As in p97–UBXD1^open, α5′ is not present in the density map, probably due to increased flexibility (Extended Data Fig. 7d). Notably, a similar rotation of the D2 small subdomain of P1 is identified in p97–UBXD1^open, supporting potential H4 occupancy (Fig. 2e). Indeed, a weak density similarly positioned atop the D2 domain was identified in the open state map at a low threshold (Extended Data Fig. 7e), indicating that H4 may be associated with the hexamer during ring opening and placing p97–UBXD1^H4 as an intermediate between the closed and open states. Based on this analysis, we predict that H4 interactions play a key role in weakening D2 interprotomer contacts, thereby driving localized opening of the D2 ring.

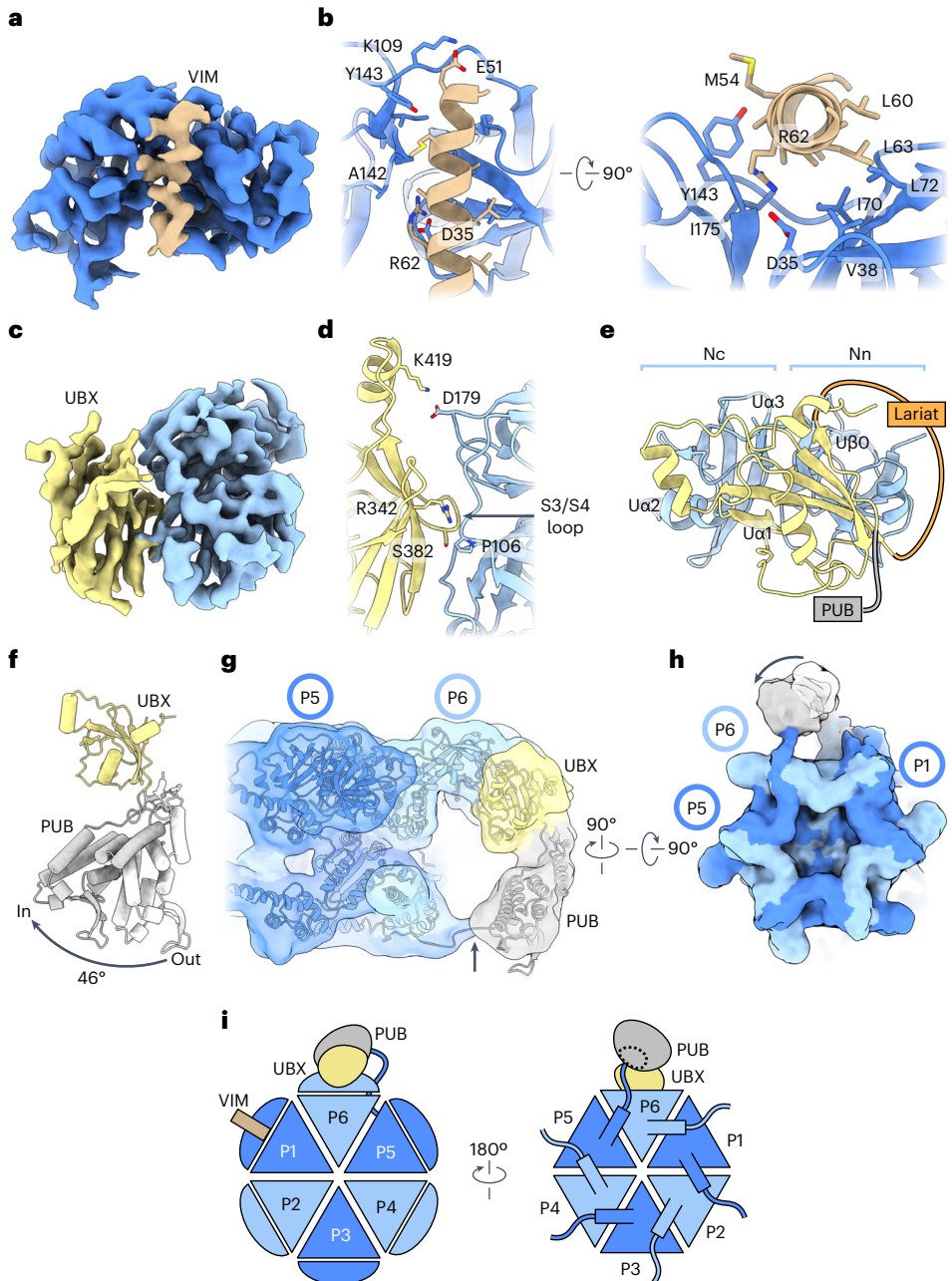

**Fig. 3 | Interactions by conserved VIM, UBX and PUB domains of UBXD1 across the p97 hexamer. a**, Sharpened map of the P1 NTD (dark blue) and the VIM helix (brown) from p97–UBXD1[closed]. **b**, Corresponding model showing VIM helix interactions with the NTD, colored as in **a**, with labeled interacting residues. **c**, Sharpened map of the P6 NTD (light blue) and UBX domain (yellow) from p97–UBXD1[closed]. **d,e**, Corresponding model of the UBX and NTD showing a conserved orientation of the S3/S4 loop[45] (arrow) (**d**) and non-canonical structural elements Uα2, Uα3 and Uβ0 (**e**), colored as in **c**. The N-terminal (Nn) and C-terminal (Nc) lobes of the NTD are indicated. **f**, Overlay of PUB domains from p97–UBXD1-PUB[in] (gray) and p97–UBXD1-PUB[out] (white), aligned to the UBX (yellow) domain, showing 46° rotation of the PUB domain position. **g,h**, Low-pass filtered map and model of p97–UBXD1-PUB[in] depicting PUB domain contact with p97 and model for C-terminal HbYX tail interaction from the adjacent P5 protomer (arrow) (**g**) and bottom view of the hexamer map with 'out' (white) and 'in' (gray) positions of the PUB (**h**). **i**, Cartoon of p97–UBXD1[closed] depicting UBXD1 interactions across three p97 protomers (P1–VIM, P6–UBX and P5–PUB) through canonical p97-interacting domains.

## The helical lariat and H4 are conserved p97-remodeling motifs

Given the rearrangements of the p97 hexamer driven by UBXD1, efforts were undertaken to identify other p97 adaptors with the UBX–helical lariat or VIM-H4 motifs. To this end, Dali searches[39] against all structures in the Protein Data Bank and the AlphaFold database were performed, revealing one protein, ASPL (also called TUG or UBXD9), with a highly similar UBX–helical lariat arrangement (Fig. 5a and Extended Data Fig. 8a). Comparison of the p97–UBXD1 structures determined here to structures of an ASPL truncation (ASPL-C) bound to p97 (ref. 33) reveals a conserved interaction (Fig. 5b). Intriguingly, ASPL also inhibits p97 ATPase activity and disassembles p97 hexamers into smaller oligomers and monomers[33,40]. Although we find no evidence of a similar hexamer disruption in UBXD1 based on our SEC or cryo-EM analysis (Extended Data Fig. 1a–d), the split ring of the p97–UBXD1[open] structure is compelling as a related function of the UBX–helical lariat in the context of UBXD1 with its additional p97 binding domains.

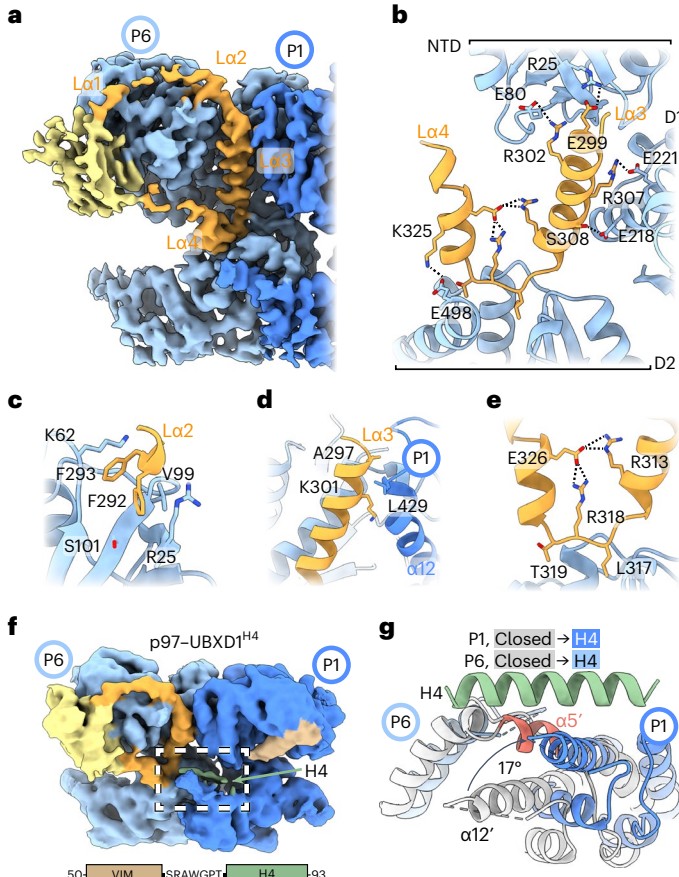

**Fig. 4 | p97 remodeling interactions by UBXD1 helical lariat and VIM–H4.**
**a**, Closed state map (from p97–UBXD1^meta) showing density for the UBXD1 helical lariat (orange) and UBX (yellow) encircling the P6 NTD with Lα2, Lα3 and Lα4 interacting along the P6–P1 interprotomer interface. **b**, Expanded view showing Lα3 and Lα4 (orange) contacts with P6 across the NTD, D1 and D2, including putative electrostatic interactions (dashed lines). **c**, View of Lα2 interactions involving hydrophobic packing into the NTD using F292 and F293. **d**, View of the P6–P1 interface showing key contacts by Lα3 with the D1 α12 helix of protomer P1. **e**, View of Lα3 and Lα4 intra-lariat contacts (between R313, R318 and E326) and contacts with D2 (by L317 and T319), stabilizing the helical lariat. **f**, Unsharpened map of p97–UBXD1^H4, showing density for H4 (green) adjacent to the VIM (brown) and along the P6–P1 interface. Shown below is an expanded view of the VIM–H4 sequence, featuring only a short, seven-amino acid linker connecting the two helices. **g**, Modeled view (see Extended Data Fig. 7c) of helix H4 interacting across the D2 domains at the P1–P6 interface from p97–UBXD1^H4 (P1, dark blue; P6, light blue), overlaid (by alignment of the P1 D2 large subdomain) with p97–UBXD1^closed (gray, showing conformational changes at the P6–P1 interface including displacement of P6 helix α5′ (red) and large rotation of P1 α12′).

Dali searches using the VIM–H4 motif did not produce any meaningful hits, probably owing to the structural simplicity of this region. However, examination of the predicted structures of other adaptors for unannotated structural elements proximal to a VIM suggested that the p97 adaptor small VCP/p97-interacting protein (SVIP) harbors an additional helix with modest similarity to UBXD1 H4 (Fig. 5c and Extended Data Fig. 8b). Given the predicted structural similarity to UBXD1 and inhibition of specific cellular functions, we reasoned that SVIP may similarly remodel p97 D2 contacts and inhibit ATPase activity.

We next purified ASPL-C[33] and SVIP and analyzed their effect on p97 ATPase activity. ASPL-C potently inhibits p97 ATPase activity, with an IC$_{50}$ of 96 nM (Fig. 5d). The complete loss of activity at high ASPL-C concentrations and highly cooperative inhibition (Hill slope, ~3) are probably a consequence of hexamer disassembly[33]. SVIP also

strongly inhibits p97 ATPase activity (IC$_{50}$ = 67 nM), indicating that the predicted helix C-terminal to the VIM contributes to ATPase inhibition through additional interactions with p97. Based on our structures and comparison to ASPL and SVIP, we postulate that the helical lariat and the H4 helix function as noncanonical control elements that, when paired with well-conserved binding motifs such as UBX and VIM, serve critical functions in ATPase control and p97 remodeling.

## Multi-domain interactions by UBXD1 confer potent ATPase inhibition

We next sought to investigate the contribution of individual UBXD1 domains on the inhibition of p97 ATPase activity. We focused on the helical lariat and H4, given their proximity to the p97 hexamer seam, and mutated them individually and in combination (Extended Data Fig. 9a and Methods). These constructs bound p97 to a similar extent as did wild-type UBXD1 (Extended Data Fig. 9b,c) and were only modestly impaired in ATPase inhibition, indicating that disruption of these elements alone does not abolish ATPase inhibition (Fig. 6a,b,d). Curiously, the major effect of the H4 mutation (in both single-mutant and double-mutant contexts) was a ~60% increase in p97 ATPase activity at maximal inhibition, indicating a substantial loss in maximal ATPase inhibition (Fig. 6b, dashed lines). These results indicate that the lariat and H4 together contribute to p97 ATPase control, although other domains are probably also operative.

We next investigated whether ATPase inhibition could be more severely impaired by simultaneously removing multiple UBXD1 domains. To that end, we purified a previously characterized N-terminal truncation construct, UBXD1-N (residues 1–133) (refs. 24,25,29) containing only the H1/H2, VIM and H4 domains, as well as a construct termed UBXD1-C (residues 94–end) with only the PUB, lariat and UBX domains (Fig. 6a). p97 ATPase inhibition was dramatically worsened in the presence of these constructs (UBXD1-C IC$_{50}$, 606 nM; UBXD1-N IC$_{50}$, not determined), indicating that interactions made by both halves of UBXD1 are necessary for potent inhibition (Fig. 6c,d). Furthermore, incubation with both UBXD1-N and UBXD1-C did not restore potency, suggesting that interactions by a single UBXD1 molecule tethered across multiple protomers are necessary.

Finally, cryo-EM analysis of p97 incubated with ADP and the UBXD1 lariat and H4 mutants was performed to understand structural changes associated with these motifs (Fig. 7a–d, Extended Data Fig. 9d–l and Table 1). Density for VIM–H4 was identified in the lariat mutant structure (p97–UBXD1^LX), and for the VIM, PUB, lariat and UBX in the H4 mutant structure (p97–UBXD1^H4X, essentially identical to p97–UBXD-1^closed). The open-ring state of p97 was not observed in either dataset, indicating that interactions by the lariat and H4 are necessary for complete separation of the P1–P6 interface. In p97–UBXD1^LX, the D2 domain, clockwise from the VIM–H4-bound protomer, exhibits strikingly weak density (Extended Data Fig. 9k), probably because mutation of the lariat also results in the loss of UBX interactions, thereby localizing UBXD1-induced conformational changes to the D2 ring of p97. To explore this state further, 3D variability analysis (Methods) was performed, revealing a variability mode in which H4 binding is correlated with destabilization of the adjacent D2 (Fig. 7c and Supplementary Video 5). Thus, these structures further demonstrate that VIM–H4 binding destabilizes p97 through disruption of D2 interprotomer contacts and, together with our ATPase analysis, support a role for the VIM–H4 interaction in D2 hydrolysis control.

We next sought to characterize distances between the seam protomers in all ADP-bound structures to further define their asymmetry and UBXD1-mediated remodeling. This was achieved by measuring distances between individual AAA+ domains (Extended Data Fig. 9m). As expected, the symmetric p97^ADP state exhibits the shortest distances (~35 Å), whereas p97–UBXD1^closed and p97–UBXD1^open show a partial (D1, ~37 Å; D2, ~36 Å) and greatly expanded (D1, ~43 Å; D2, ~46 Å) separation of the AAA+ domains, respectively (Fig. 7d). As shown above

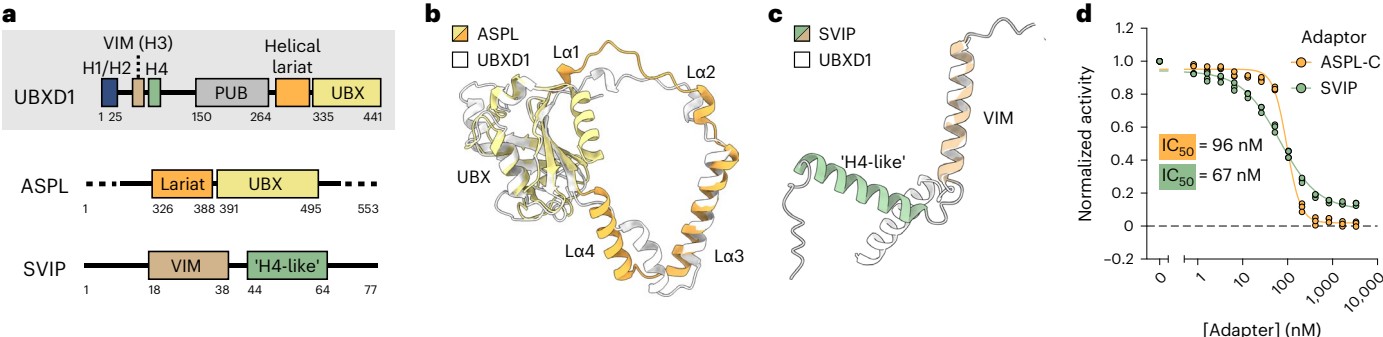

**Fig. 5 | Analysis of the helical lariat and VIM-H4 as conserved p97-remodeling motifs. a**, Domain schematics of UBXD1, ASPL and SVIP (not to scale). **b,c**, Overlay of the UBX-helical lariat of ASPL (residues 318-495 from PDB 5IFS, colored as in **a**) and UBXD1 (residues 270-441 from the p97-UBXD1$^{closed}$ model, in white) (**b**) and the VIM-'H4-like' region of SVIP (AlphaFold model, colored as in **a**) and UBXD1 (residues 50-93 from the AlphaFold model, in white) (**c**). **d**, Steady-state ATPase activity of p97 as a function of ASPL-C or SVIP concentration (normalized to activity at 0 nM adaptor). Data are from $n = 3$ independent experiments, each with three technical replicates. Data are presented as mean values from each independent experiment. Calculated IC$_{50}$ values are also shown.

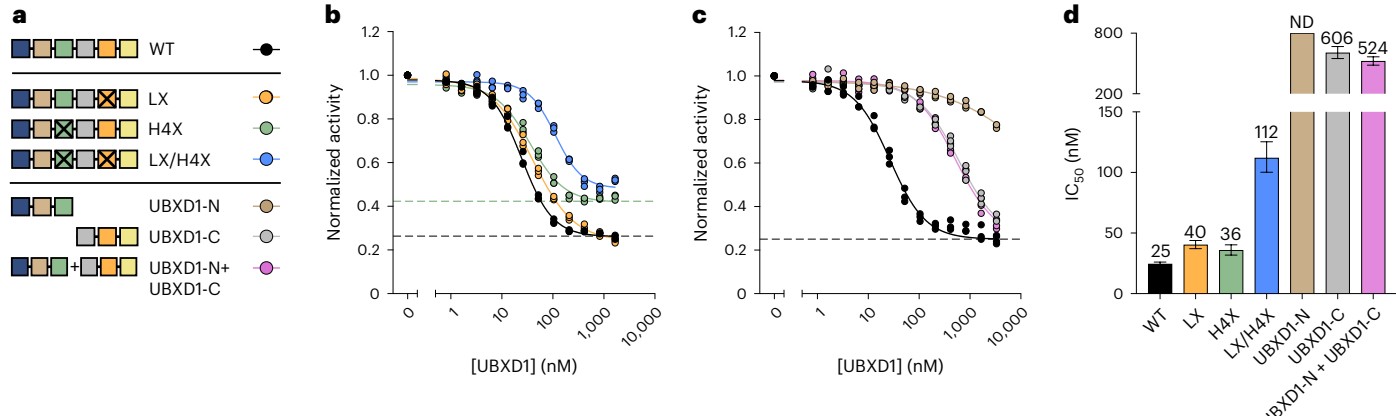

**Fig. 6 | Mutational analysis of UBXD1. a**, Schematic of UBXD1 mutants tested. **b**, Steady-state ATPase activity of p97 as a function of UBXD1 protein concentration for WT (wild type), LX (lariat mutant: E299R/R302E/R307E/E312R) or H4X (helix H4 sequence scramble) (normalized to activity at 0 nM UBXD1). Dashed lines represent the minimal activity (or maximal UBXD1 inhibition) obtained from the corresponding curve fit. Data are from $n = 3$ independent experiments, each with three technical replicates. Data are presented as mean values from each independent experiment. **c**, As in **b**, but for UBXD1-N (residues 1-133), UBXD1-C (residues 94-end) or an equimolar combination of UBXD1-N and UBXD1-C. Data are from $n = 3$ independent experiments, each with three technical replicates. Data are presented as mean values from each independent experiment. **d** Calculated IC$_{50}$ values for ATPase inhibition by UBXD1 mutants. Data are from $n = 3$ independent experiments, each with three technical replicates. Error bars, 95% CI; ND, not determined.

(Fig. 2a), in p97-UBXD1$^{closed}$, the D1 domains are more separated than the D2 domains, probably owing to lariat binding. Mutation of the lariat decreases the D1-D1 distance relative to p97-UBXD1$^{closed}$, and mutation of H4 shows no changes in D1-D1, further supporting that the D1 remodeling effects are driven by the lariat interaction. Intriguingly, the D2-D2 distance in p97-UBXD1$^{LX}$ is substantially increased to ~40 Å relative to p97-UBXD1$^{closed}$ (at ~36 Å). We postulate that this reflects a VIM-H4 interaction that is more pronounced when uncoupled from lariat binding to the D1 (as observed in Fig. 7c), further suggesting that VIM-H4 interactions indeed contribute to ring opening through D2 displacement. In summary, these results indicate that the helical lariat and H4 independently regulate the D1 and D2 domains and function as critical p97 hexamer-disrupting motifs that are necessary for full UBXD1 remodeling activity.

## Discussion

Adaptor proteins of p97/VCP serve critical roles in regulating the function of its many diverse and essential cellular pathways. How adaptors directly regulate p97 structure and mechanism has been an open question. Here, we characterize the multi-domain adaptor UBXD1, which is associated with lysosomal and mitochondrial clearance, among other functions[20–24]. We identify UBXD1 as a potent ATPase inhibitor and determine structures of full-length p97-UBXD1 that reveal how its interactions drive the dramatic remodeling and ring-opening of the hexamer. These conformational changes are coordinated by UBXD1 through an extensive network of interprotomer interactions across the NTD, D1 and D2, which is probably necessary owing to the stability of the p97 hexamer[10,41]. Based on these structures, we propose a model describing how UBXD1 interactions coalesce to remodel p97 (Fig. 7e,f and Supplementary Video 6).

We identify VIM binding to the NTD to be a primary contact given its established interaction[24,27,29,30] and the prevalence of the p97-UBXD1$^{VIM}$ state, but this interaction is insufficient to induce remodeling (Fig. 7e, state II and Extended Data Fig. 1e). Rather, interactions made by the helical lariat, including strong contacts by Lα3 along the D1 interprotomer interface, drive remodeling and separation of adjacent D1 domains (Fig. 7e, state III). This separation is supported by UBX binding to the clockwise NTD, given its connection to the lariat. Through subclassification of the closed state, we identify a class with the UBXD1 H4 helix, which is connected to the VIM by a short linker, bound at the

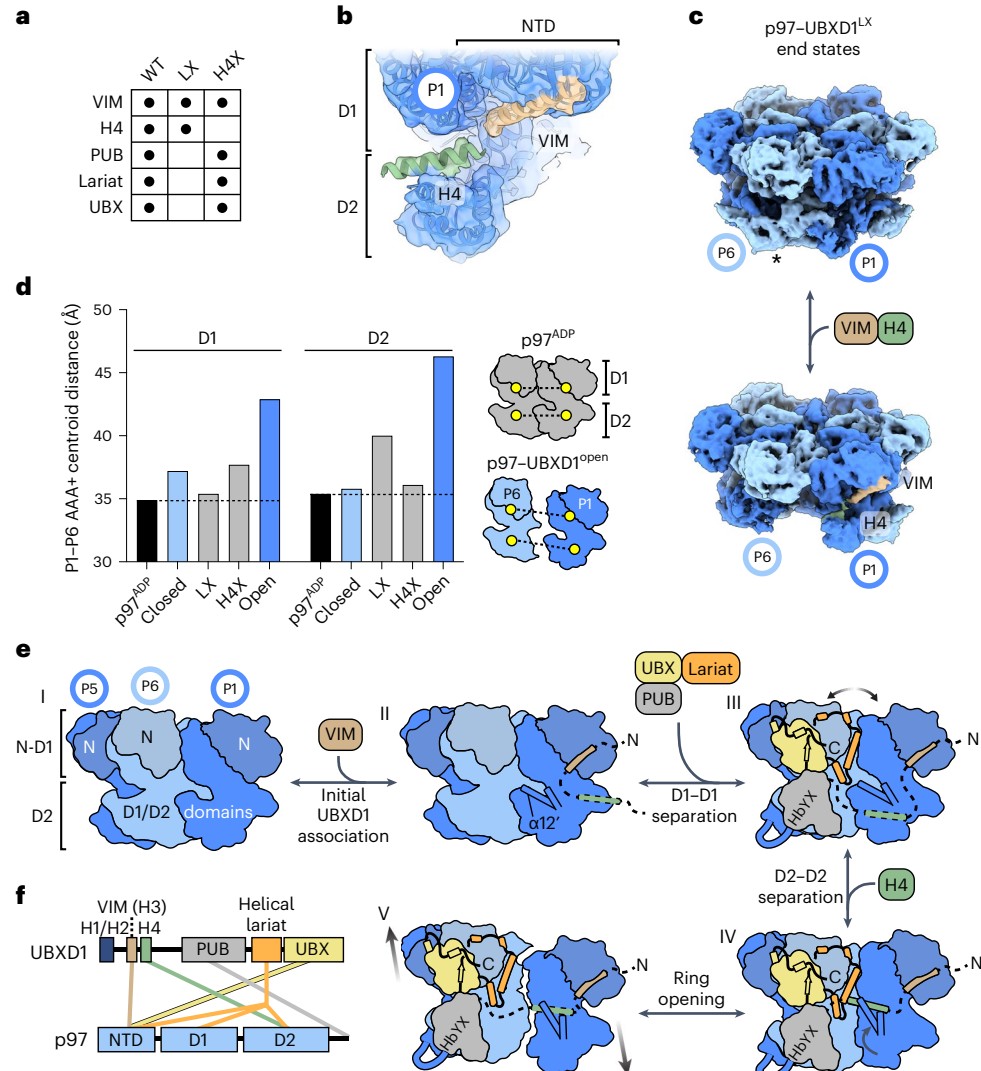

**Fig. 7 | Structural analysis of p97–UBXD1 mutant complexes and model for p97 hexamer remodeling through UBXD1 domain interactions. a**, Table of ADP-bound p97–UBXD1 cryo-EM datasets (WT, LX and H4X) and corresponding UBXD1 domains observed as densities in the reconstructions. **b**, Unsharpened map and fitted model of the VIM–H4-bound P1 protomer from p97–UBXD1$^{LX}$ (Extended Data Fig. 9d), colored as in Fig. 1. **c**, First and last frames of the 3D variability analysis output for p97–UBXD1$^{LX}$ showing P6 D2 density (*) but no VIM–H4 in one end state (top) and no P6 D2 in the other when VIM–H4 density is present (bottom). **d**, P1–P6 interprotomer distances (based on centroid positions) for the D1 and D2 domains of p97$^{ADP}$ (PDB 5FTK), p97–UBXD1$^{closed}$, p97–UBXD1$^{LX}$, p97–UBXD1$^{H4X}$ and p97–UBXD1$^{open}$. Dashed lines represent the minimal distances observed in p97$^{ADP}$. A schematic representing the distances calculated is shown

(right). **e**, Model of p97–UBXD1 interactions and structural remodeling of the hexamer. State I, side view of p97$^{ADP}$ (PDB 5FTK), colored as in Fig. 1. NTDs are shaded for clarity. State II, p97–UBXD1$^{VIM}$, in which the VIM initially associates with the NTD of P1. The position of the D2 small subdomain is illustrated by α12′ and an adjacent helix. State III, the p97–UBXD1$^{closed}$ state, in which the UBX, PUB and helical lariat contact P1, P5 and P6, resulting in the disruption of D1–D1 contacts at the P1–P6 interface. State IV, the p97–UBXD1$^{H4}$ state, in which H4 is positioned on top of the D2 domain of P1, causing it to rotate upward and displacing a helix from the D2 domain of P6. State V, the p97–UBXD1$^{open}$ state, in which P6 and P1 have completely separated and all protomers are arranged into a shallow right-handed helix. **f**, Summary of p97–UBXD1 interactions identified in this study.

D2 interface (Fig. 4f). This interaction appears to be critical for separation of the D2 domains, given conformational changes identified in this state that destabilize the interface, including displacement of α5′ (Fig. 4g). Moreover, variability analysis of p97–UBXD1$^{LX}$ reveals that H4 interactions are dynamic and displace the adjacent D2 (of protomer P6), further supporting a direct role in disrupting the D2 interface (Fig. 7c). Therefore, we propose that VIM–H4 specifically functions in p97 D2 remodeling while the combined interactions from the lariat and H4 together drive hexamer opening (Fig. 7e, state V). The connecting UBX and VIM domains tether these interactions to the respective NTDs, providing additional binding energy to leverage ring opening. Although more flexible, the PUB interaction with the C terminus of the

next clockwise protomer may further support D1–D2 remodeling and ATPase control (Fig. 6).

Notably, while other UBXD1-bound configurations are identified (Extended Data Fig. 3), the open state is only observed with a singly bound, wild-type UBXD1 in the ADP state, indicating that all UBXD1 contacts across three protomers are required for ring opening. Additional UBXD1 molecules may transiently bind p97; however, opening of the hexamer at one site would probably displace other molecules because of steric interactions and conformational changes across the ring. The absence of an open state in the presence of ATPγS may be caused by increased interprotomer interactions and hexamer stability. Indeed, in substrate-bound AAA+ complexes, hydrolysis at the spiral

seam is thought to destabilize the interprotomer interface, facilitating substrate release and rebinding during stepwise translocation[42].

The conformational changes between the symmetric (p97[ADP]), closed and open states (Supplementary Video 2) appear connected and on-path to the right-handed spiral adopted by the substrate-bound state[6,7,11], wherein protomer interfaces at the spiral seam are destabilized during translocation steps. Therefore, we propose that UBXD1 may bind and remodel p97 to support substrate loading or cycles of translocation (Fig. 2d). Indeed, UBXD1 colocalizes with p97, YOD1 (a deubiquitinase) and PLAA (a ubiquitin-binding adaptor) in lysophagy[21], and the UBXD1 UBX domain has been reported to bind ubiquitin (and potentially thus ubiquitylated substrates)[32]. Together, these results indicate that UBXD1 may be directly involved in regulating p97 substrate processing.

UBXD1's potent inhibition of p97 ATPase activity is striking and indicates a distinct role for adaptors in regulating hydrolysis. Inhibition may result from disruption of D1 and D2 interprotomer contacts, given that the intersubunit signaling contacts from the adjacent protomer are lost (Extended Data Fig. 4d,e and ref. 11). Although we consider the helical lariat and H4 to be primary drivers of ring separation and thus ATPase inhibition, the effects of these remodeling elements are probably buttressed by other UBXD1 domains, given that mutation of the lariat or H4 does not completely ablate UBXD1's inhibitory activity but removing multiple domains at once has stronger effects (Fig. 6b–d). Therefore, the potent inhibitory effect of UBXD1 may be driven by avidity of multiple interactions, without a single domain being responsible for hydrolysis control. Supporting this idea, many UBXD1 interactions are weak or weakened compared to homologous domains in other adaptors. Specifically, H1/H2 and H4 are highly conserved (Extended Data Fig. 10) but weakly interact by nuclear magnetic resonance spectroscopy[24–26,43], the VIM lacks an arginine residue that is present in many other adaptors[29,30] and the UBX has S3/S4 loop mutations and does not bind the NTD in isolation[20,27,31]. Notably, UBXD1-mediated inhibition of ATPase activity does not necessitate an overall inhibitory role; UBXD1 could potentially stabilize a stalled, ATPase-inhibited state until substrate association, at which point hydrolysis-dependent substrate processing might occur.

We propose that UBXD1, ASPL and SVIP are structurally related adaptors with conserved motifs that mediate distinct effects on p97 activity. In addition to being potent inhibitors of p97 ATPase activity, our results suggest that the primary activity of these adaptors is the modulation of p97 structure, rather than direct involvement in substrate engagement, given their lack of conserved substrate-binding domains. However, a recent study[32] reporting a ubiquitin-binding ability of the UBXD1 UBX domain may indicate that this adaptor is indeed involved in substrate binding. We identify the helical lariat and H4-like sequences to be critical control elements in these adaptors. The different degrees of remodeling of lariat-containing adaptors (hexamer disassembly with ASPL compared to intact hexamers with UBXD1) may reflect the different assembles of p97-interacting domains in these two adaptors. Moreover, a recent study revealed that ASPL-mediated hexamer disassembly enables binding and modification by the methyltransferase VCPKMT[44], suggesting that disruptions of hexamer architecture are biologically relevant. To summarize, the characterization of adaptors as structural modulators of p97 reported here and the large number of still-uncharacterized p97-interacting proteins suggest that there are more classes of adaptors with distinct effects on p97 unfoldase function yet to be discovered.

## Online content

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

## Methods

### Molecular cloning

The coding sequence of full-length human UBXD1 was cloned with an N-terminal 6×His tag, maltose-binding protein (MBP) tag and tobacco etch virus (TEV) protease cleavage site into an insect cell expression vector (Addgene, plasmid no. 55218) and a bacterial expression vector (Addgene, plasmid no. 29708). ASPL-C (residues 313–553) and full-length SVIP were cloned into the same bacterial expression vector (including the 6×His–MBP–TEV tag). The Q5 Site-Directed Mutagenesis Kit (New England BioLabs) was used to introduce mutations into the UBXD1 construct. The UBXD1 lariat was mutated by making four charge reversals in Lα3 that were predicted to disrupt contacts with P1 and P6 (E299R/R302E/R307E/E312R), given that this helix makes the most substantial contacts with p97. As we could not obtain high-resolution structural information about H4, we scrambled the sequence of this helix rather than making point mutants, using Peptide Nexus Sequence Scrambler (https://peptidenexus.com/article/sequence-scrambler). This resulted in the sequence 75-QSRDVTQERIQNKAVLTEA-93. The expression vectors for UBXD1-N (containing residues 1–133 of UBXD1) and UBXD1-C (residues 94–end) were generated by deleting extraneous regions from the wild-type bacterial expression vector. Primers used to generate the above constructs can be found in Supplementary Table 1.

### Protein expression and purification

p97 was expressed and purified as previously described[46]. In brief, BL21-Gold(DE3) chemically competent *E. coli* (Agilent) were transformed with pET15b p97, encoding full-length p97 with an N-terminal 6×His tag, grown in 2xYT media supplemented with 100 µg ml$^{-1}$ carbenicillin and induced with 0.5 mM isopropyl β-D-1-thiogalactopyranoside (IPTG) at 20 °C overnight. Cells were collected and lysed by sonication in lysis buffer (50 mM Tris pH 8.0, 250 mM NaCl, 10 mM imidazole, 0.5 mM Tris(2-carboxyethyl)phosphine (TCEP), 1 mM phenylmethylsulfonyl fluoride (PMSF)) supplemented with cOmplete EDTA-free Protease Inhibitor Cocktail (Roche) and then clarified by centrifugation. The supernatant was then incubated with HisPur Ni-NTA resin (Thermo Scientific), and p97 was eluted with nickel elution buffer (lysis buffer supplemented with 320 mM imidazole, no PMSF). The eluate was supplemented with TEV protease and dialyzed overnight at 4 °C into p97 dialysis buffer (10 mM Tris pH 8.0, 100 mM NaCl, 1 mM dithiothreitol). The following day, the cleavage product was passed through fresh Ni-NTA resin, and the flowthrough was concentrated and applied to a HiLoad 16/600 Superdex 200 pg SEC column (GE Healthcare) equilibrated in p97 SEC buffer (25 mM HEPES pH 7.4, 150 mM KCl, 5 mM MgCl$_2$, 0.5 mM TCEP). Fractions containing p97 were concentrated to >200 µM, filtered and flash-frozen in liquid nitrogen.

Initial attempts at expression of UBXD1 in *E. coli* resulted in large amounts of insoluble material. Therefore, to obtain amounts sufficient for initial studies, a UBXD1 construct with a TEV-cleavable N-terminal 6×His–MBP tag was expressed in Sf9 insect cells (Expression Systems, no. 94-001F) using standard methods. This protocol (including cleavage of the 6×His–MBP tag) yielded sufficient material for preliminary cryo-EM studies and was used for the p97–UBXD1$^{WT}$•ADP superstoichiometric dataset. Thereafter, an optimization campaign for soluble expression of 6×His–MBP–UBXD1 in *E. coli* was performed, which resulted in the following protocol. *E. coli*-derived UBXD1 and mutants thereof were used for all biochemical experiments, and the p97–UBXD1•ATPγS, p97–UBXD1•ADP substoichiometric, p97–UBXD1$^{LX}$•ADP and p97–UBXD1$^{H4X}$•ADP datasets. BL21-Gold(DE3) chemically competent *E. coli* (Agilent) were transformed with the UBXD1 expression vectors and used for large-scale expression. Cells were grown in 2xYT media supplemented with 100 µg ml$^{-1}$ carbenicillin at 37 °C until OD$_{600}$ reached ~1.25, then protein expression was induced with 0.5 mM IPTG and grown for 1 h at 37 °C. Cells were rapidly cooled in an ice bath for 10 min, then collected by centrifugation at 10,000×*g* and stored at

−80 °C until use. All subsequent steps were performed at 4 °C. Pellets were resuspended in lysis buffer (see above) supplemented with cOmplete EDTA-free Protease Inhibitor Cocktail (Roche) and lysed by sonication. Lysates were clarified by centrifugation at 85,000×*g* and incubated with HisPur Ni-NTA resin (Thermo Scientific) for 15 min. The resin was washed with nickel wash buffer (lysis buffer without imidazole or PMSF) and eluted with nickel elution buffer (see above). The eluate was concentrated, filtered and applied to a HiLoad 16/600 Superdex 200 pg SEC column (GE Healthcare) equilibrated in adaptor SEC buffer (25 mM HEPES pH 7.4, 150 mM KCl, 5% glycerol (v/v), 0.5 mM TCEP). TEV protease was added to fractions containing MBP–UBXD1 and incubated overnight without agitation. The following day, the sample was passed through a 5 ml MBPTrap HP column (GE Healthcare) to remove the cleaved 6×His–MBP tag, and the flowthrough was concentrated before 15× dilution with anion exchange (AEX) binding buffer (25 mM HEPES pH 7.5, 0.5 mM TCEP). The diluted sample was then applied to a 1 ml HiTrap Q HP column (GE Healthcare), and UBXD1 was eluted with a 0–50% gradient of AEX elution buffer (AEX binding buffer supplemented with 1,000 mM KCl), concentrated to >200 µM and flash-frozen in liquid nitrogen.

To express ASPL-C and SVIP, BL21-Gold(DE3) chemically competent *E. coli* (Agilent) were transformed with pET MBP–ASPL-C and pET MBP–SVIP and grown in Terrific Broth supplemented with 100 µg ml$^{-1}$ ampicillin at 37 °C until OD$_{600}$ reached ~2, then protein expression was induced with 0.4 mM IPTG and grown overnight at 18 °C. Cells were collected by centrifugation at 4,000×*g* and stored at −80 °C or processed immediately. All subsequent steps were performed at 4 °C. Cell lysis and nickel immobilized metal affinity chromatography were performed as for UBXD1. TEV protease was added to the eluates, and the solutions were dialyzed overnight in adaptor dialysis buffer (25 mM Tris pH 8.0, 150 mM NaCl, 0.5 mM TCEP). The following day, the samples were passed through fresh Ni-NTA resin to remove the 6×His–MBP tags, concentrated and applied to a HiLoad 16/600 Superdex 200 pg SEC column (GE Healthcare) equilibrated in adaptor SEC buffer. Fractions containing adaptor proteins were concentrated to >200 µM and flash-frozen in liquid nitrogen.

Purity of all proteins was verified by SDS–PAGE and concentration was determined using the Pierce BCA Protein Assay Kit (Thermo Scientific).

### ATPase assays

The ATPase assay protocol was modified from previously published methods[47]. In an untreated 384-well microplate (Grenier 781101), 50 µl solutions were prepared with a final concentration of 10 nM p97 hexamer, variable adaptor (UBXD1 and mutants, ASPL-C and SVIP) and 200 µM ATP (Thermo Fisher Scientific) in ATPase buffer (25 mM HEPES pH 7.4, 100 mM KCl, 3 mM MgCl$_2$, 1 mM TCEP, 0.1 mg ml$^{-1}$ BSA). For experiments with 1:1 mixtures of UBXD1-N and UBXD1-C, concentration refers to each construct individually (that is, 1 µM = 1 µM UBXD1-N and 1 µM UBXD1-C). ATP was added last to initiate the reaction, and the solutions were incubated at room temperature (22 °C) until 8% substrate hydrolysis was achieved. To quench the reaction, 50 µl of BIOMOL Green (Enzo Life Sciences) was added and allowed to develop at room temperature for 25 min before reading at 620 nm. Data were fit to the model for [inhibitor] vs. response–variable slope (four parameters) in Prism v.9.3.1 (GraphPad). IC$_{50}$ values were determined using the equation:

$$Y = Bottom + \frac{Top - Bottom}{1 + \left(\frac{IC_{50}}{X}\right)^{HillSlope}}$$

### Analytical SEC

For SEC analysis, 60 µl samples (10 µM p97 monomer, 20 µM UBXD1 and 5 mM nucleotide where applicable) were prepared in p97 SEC buffer

and incubated on ice for 10 min. Samples were filtered and injected on a Superose 6 Increase 3.2/300 column (GE Healthcare) equilibrated in p97 SEC buffer and operated at 8 °C. Fractions of 100 µl were collected and analyzed by SDS–PAGE with Coomassie Brilliant Blue R-250 staining (Bio-Rad).

## Cryo-EM sample preparation, data collection and image processing

For all p97–UBXD1 datasets except p97–UBXD1•ADP substoichiometric, 10 µM p97 monomer and 20 µM UBXD1 were incubated with 5 mM ADP or ATPγS in p97 SEC buffer for 10 min on ice before vitrification. For the substoichiometric dataset, 10 µM and 1.67 µM were used, respectively. A 3 µl drop was applied to a glow-discharged (PELCO easiGlow, 15 mA, 2 min) holey carbon grid (Quantifoil R1.2/1.3 on gold 200 mesh support), blotted for 3–4 s with Whatman Grade 595 filter paper (GE Healthcare) and plunge-frozen into liquid ethane cooled by liquid nitrogen using a Vitrobot (Thermo Fisher Scientific) operated at 4 °C and 100% humidity. Samples were imaged on a Titan Krios TEM (Thermo Fisher Scientific) operated at 300 kV and equipped with a Bio-Quantum K3 Imaging Filter (Gatan) using a 20 eV zero loss energy slit. Movies were acquired with SerialEM 4.1.0 (ref. [48]) in super-resolution (UBXD1$^{WT}$•ADP, UBXD1$^{WT}$•ATPγS, UBXD1$^{WT}$•ADP substoichiometric and UBXD1$^{H4X}$•ADP) or counted (UBXD1$^{LX}$•ADP) mode at a calibrated magnification of ×59,952, corresponding to a physical pixel size of 0.834 Å. A nominal defocus range of −0.8 µm to −1.8 µm was used with a total exposure time of 2 s fractionated into 0.255 s frames for a total dose of 43 e$^-$/Å$^2$ at a dose rate of 15 e$^-$ per pixel per second. Movies were subsequently corrected for drift and dose-weighted using MotionCor2 v1.6.4 (ref. [49]), and the micrographs collected in super-resolution mode were Fourier cropped by a factor of two.

For the p97–UBXD1$^{WT}$•ADP sample, a total of 22,536 micrographs were collected and initially processed in cryoSPARC v3.3 (ref. [50]). After Patch CTF estimation, micrographs were manually curated to exclude those of poor quality, followed by blob-based particle picking, 2D classification, ab initio modeling and initial 3D classification. Three classes of interest from the initial 3D classification were identified, corresponding to a closed-like state (class 1), an open-like state (class 2) and p97–UBXD1$^{VIM}$ (class 3). For p97–UBXD1$^{VIM}$, 100,000 particles were randomly selected and refined with C6 symmetry imposed. For the open state, refinement of all particles produced a map with poor resolution for protomers P1 and P6, so focused classification without image alignment (skip-align) of these individual protomers was performed in RELION 3.1 (ref. [51]), followed by masked local refinement in cryoSPARC. A composite map of these two local refinements and protomers P2–P5 from the consensus map was generated in UCSF Chimera 1.16 (ref. [52]) by docking the local refinement maps into the consensus map, zoning each map by a radius of 4 Å using the associated chains from a preliminary model and summing the aligned volumes. For the closed state, refinement of the closed-like particles revealed additional UBXD1 density at protomers other than P1 and P6, so skip-align focused classification using a mask encompassing protomers with additional density was performed in RELION, followed by homogenous refinement in cryoSPARC, yielding the p97–UBXD1$^{meta}$ and p97–UBXD1$^{para}$ states. The class from focused classification corresponding to a singly bound hexamer was also refined in cryoSPARC and then subjected to skip-align focused classification of P1 and P6 in RELION, followed by homogenous refinement in cryoSPARC. This yielded the p97–UBXD1$^{closed}$ and p97–UBXD1$^{H4}$ states. To obtain better density for the PUB domain, all particles from the closed-like class from the initial 3D classification were used for skip-align focused classification of the PUB and UBX domain in RELION, followed by homogenous refinement in cryoSPARC. This yielded p97–UBXD1-PUB$_{out}$ and p97–UBXD1-PUB$_{in}$. To visualize variability in this dataset, 3D variability analysis in cryoSPARC was performed using particles from the closed-like and open-like states.

For the p97–UBXD1$^{WT}$•ATPγS sample, a total of 9,498 micrographs were collected and processed as for p97–UBXD1$^{WT}$•ADP. An initial 3D classification revealed three classes of interest: a state resembling p97–UBXD1$^{closed}$, a state resembling a fully ATPγS-bound p97 hexamer with NTDs in the 'up' state and a state resembling the p97–UBXD1$^{VIM}$ state. Homogenous refinement with defocus refinement was performed for each of these classes, and the resolution in all maps was sufficient to assign nucleotide density as ATPγS in all nucleotide pockets in all protomers in all structures.

For the p97–UBXD1$^{WT}$•ADP substoichiometric sample, a total of 8,569 micrographs were collected and processed as for p97–UBXD1$^{WT}$•ADP. An initial 3D classification revealed three classes of interest: one closely resembling p97–UBXD1$^{open}$ and the other two resembling unbound p97 hexamers. No further processing was performed with these classes.

For the p97–UBXD1$^{LX}$•ADP sample, a total of 6,330 micrographs were collected and initially processed as for p97–UBXD1$^{WT}$•ADP. An initial 3D classification revealed two high-resolution classes: one featuring density for the UBXD1 VIM in all NTDs (class 1) and the other featuring stronger density for the VIM in one NTD than in the others (class 2). In this class, the D2 domain of the protomer counterclockwise from the best VIM-bound protomer had much weaker density, indicating flexibility. Homogenous refinement of class 1 produced a map essentially identical to p97–UBXD1$^{VIM}$. Homogenous refinement of class 2 produced a map with weak density putatively corresponding to H4 on the strong VIM-bound protomer, so skip-align focused classification of the D2 domain of this protomer was performed in RELION, followed by a final homogenous refinement in cryoSPARC of the best class. This yielded a map with improved VIM and H4 density. To visualize variability in this dataset, 3D variability analysis in cryoSPARC was performed using particles from the class 2 refinement (pre-focused classification).

For the p97–UBXD1$^{H4X}$•ADP sample, a total of 12,418 micrographs were collected and initially processed as for p97–UBXD1$^{WT}$•ADP. An initial 3D classification revealed three high-resolution classes: one featuring density for the UBXD1 VIM in all NTDs (class 1) and the other two (classes 2 and 3) featuring stronger density for the VIM and additional UBXD1 domains, including the PUB, helical lariat and UBX. These classes strongly resemble p97–UBXD1$^{closed}$. Homogenous refinement of class 1 produced a map essentially identical to p97–UBXD1$^{VIM}$. Homogenous refinement of classes 2 and 3 combined produced a map with density for the VIM, PUB, lariat and UBX but without H4 density or associated movements of the D2 domains. To identify particles with H4 density, skip-align focused classification of protomers P1 and P6 was performed in RELION, which did not reveal any classes with density attributable to H4.

## 3D classification of p97–UBXD1 states using common references

To quantify the abundance of distinct states across all datasets with wild-type UBXD1, heterogenous refinement in cryoSPARC was performed separately on each dataset, using all particles selected after 2D classification. The same six classes were used as starting references for all jobs (closed state, open state, VIM-only state, p97 double-ring, p97 with NTDs in the up conformation and a junk class). Classes that produced low-resolution, artifactual or otherwise poor-quality reconstructions were discarded and the proportions of the remaining classes were plotted.

## Molecular modeling

To generate the model for p97–UBXD1$^{closed}$, a model of the p97$^{ADP}$ hexamer[36] and the AlphaFold model of UBXD1 (refs. [34],[35]) were docked into the map using UCSF Chimera and ISOLDE 1.3 (ref. [53]) in UCSF ChimeraX 1.16 (ref. [54]), followed by refinement using Rosetta v3.12 Fast Torsion Relax. Models for all other structures were generated by docking individual chains (or regions thereof) from the closed model, followed by refinement using Rosetta Fast Torsion Relax. The AlphaFold model for the H4 helix was manually placed into the corresponding

density in p97–UBXD1[H4] and p97–UBXD1[LX]. Sidechains for H4 are omitted owing to low map resolution. Coot v0.8.9.2 (ref. 55), ISOLDE and Phenix v1.20.1 (ref. 56) were used to finalize all models.

### Calculation of buried surface area
The 'measure buriedarea' command with default parameters in UCSF ChimeraX was used to measure buried surface area of p97–UBXD1 interactions in the closed and open states. The area buried by the UBXD1 PUB was not considered, as the experimental density for the p97 C-terminal tail in either state was not of sufficient quality to model.

### Protein sequence alignments
Amino acid sequences were aligned in MUSCLE v5 (ref. 57) and visualized in MView v1.67 (ref. 58).

### Data analysis and figure preparation
Biochemical data were analyzed and plotted using Prism v.9.3.1 (Graph-Pad). Figures were prepared using Adobe Illustrator, UCSF Chimera and UCSF ChimeraX[52,54].

### Reporting summary
Further information on research design is available in the Nature Portfolio Reporting Summary linked to this article.

### Data availability
Cryo-EM densities have been deposited at the Electron Microscopy Data Bank under accession codes EMD-28982 (p97–UBXD1[closed]), EMD-28983 (p97–UBXD1[open] composite), EMD-28984 (p97–UBXD1[open] consensus), EMD-28985 (p97–UBXD1[open] P1 focused map), EMD-28986 (p97–UBXD1[open] P6 focused map), EMD-28987 (p97–UBXD1[VIM]), EMD-28988 (p97–UBXD1[meta]), EMD-28989 (p97–UBXD1[para]), EMD-28990 (p97–UBXD1-PUB[in]), EMD-28991 (p97–UBXD1[H4]) and EMD-28992 (p97–UBXD1[LX]). Atomic coordinates have been deposited at the Protein Data Bank under accession codes PDB 8FCL (p97–UBXD1[closed]), PDB 8FCM (p97–UBXD1[open]), PDB 8FCN (p97–UBXD1[VIM]), PDB 8FCO (p97–UBXD1[meta]), PDB 8FCP (p97–UBXD1[para]), PDB 8FCQ (p97–UBXD1-PUB[in]), PDB 8FCR (p97–UBXD1[H4]) and PDB 8FCT (p97–UBXD1[LX]). Accession codes for additional models referenced in this study are PDB 5FTK (p97[ADP]), PDB 5FTN (p97[ATPγS]), PDB 5FTJ (p97–UPCDC30245) PDB 5IFS (p97–ASPL-C), PDB 3TIW (NTD–gp78-VIM), PDB 5X4L (NTD–UBXD7–UBX), AF-Q9BZV1-F1 (UBXD1 AlphaFold model) and AF-Q8NHG7-F1 (SVIP AlphaFold model). Uncropped images for Extended Data Figs. 1b and 9c and data used for all biochemical experiments are provided as source data online. Source data are provided with this paper.

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

### Acknowledgements
This work was supported by National Instiututes of Health grants F31GM142279 (to J.R.B.), R01GM130145 (to M.R.A.) and R01GM138690 (to D.R.S.).

### Author contributions
J.R.B. cloned protein expression constructs, expressed and purified proteins, performed biochemical and cryo-EM experiments, built models, developed figures and wrote and edited the paper. C.R.A. cloned protein expression constructs, expressed and purified proteins, performed biochemical experiments and edited the paper. M.R.T. expressed and purified proteins and performed cryo-EM experiments. E.T. operated electron microscopes and assisted with data collection. A.C.T. expressed proteins. M.R.A. designed and supervised biochemistry and edited the paper. D.R.S. designed and supervised the project and wrote and edited the paper.

### Competing interests
The authors declare no competing interests.

### Additional information
**Extended data** is available for this paper at https://doi.org/10.1038/s41594-023-01126-0.

**Correspondence and requests for materials** should be addressed to Michelle R. Arkin or Daniel R. Southworth.

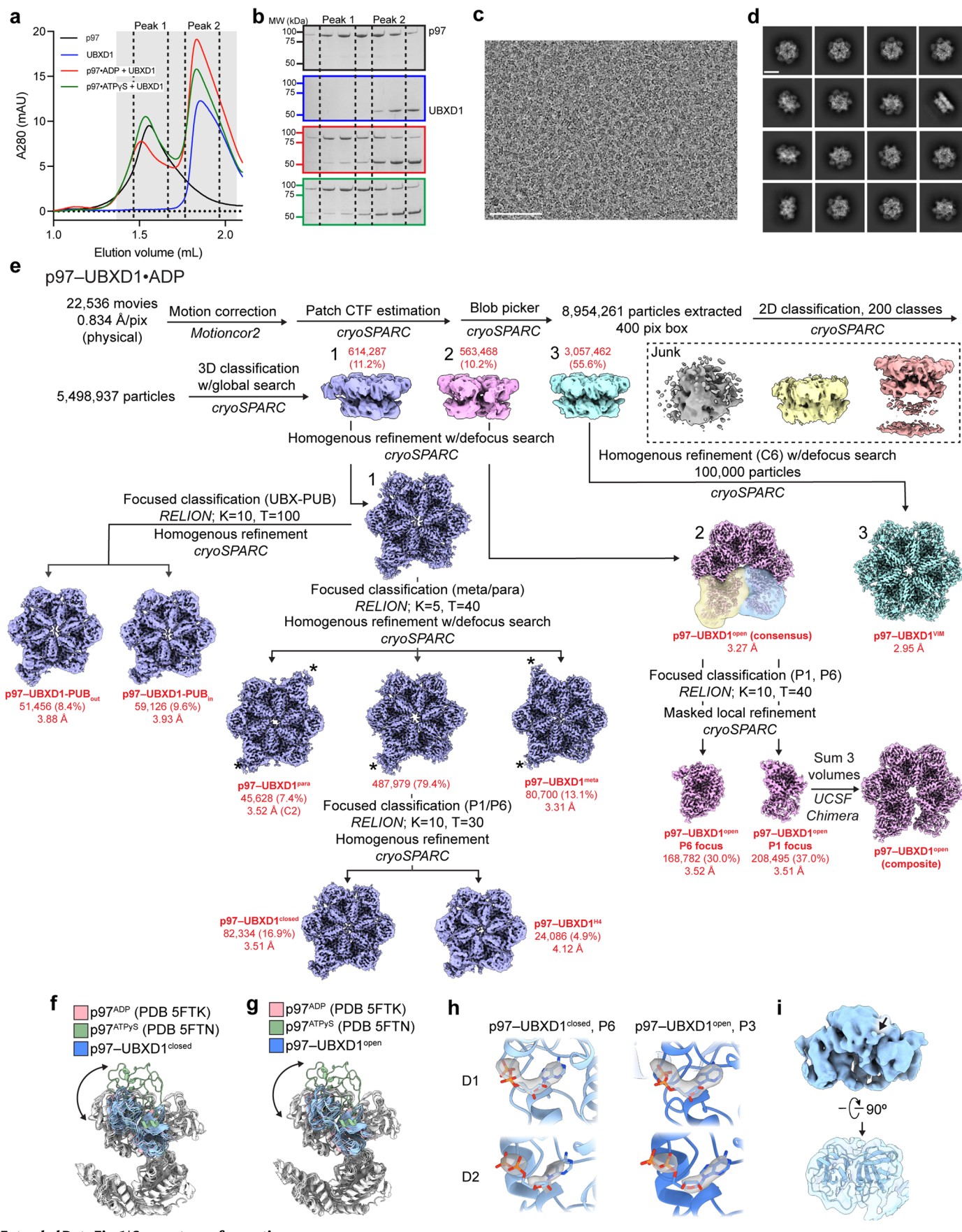

**Extended Data Fig. 1 | See next page for caption.**

**Extended Data Fig. 1 | Biochemical and cryo-EM analysis of the p97–UBXD1 interaction.** (**a**) SEC traces of p97–UBXD1 samples. Fractions in the shaded range were analyzed by SDS–PAGE. No p97 monomer peak was observed with UBXD1 incubation. Data are representative of three independently performed experiments with similar results. (**b**) Coomassie Brilliant Blue-stained SDS–PAGE gels of fractions from SEC runs in (**a**). Data are representative of three independently performed experiments with similar results. (**c**) Representative micrograph of the p97–UBXD1$^{WT}$•ADP dataset (scale bar equals 100 nm). (**d**) Representative 2D class averages of the p97–UBXD1$^{WT}$•ADP dataset (scale bar equals 100 Å). No p97 monomers were identified during 2D classification. (**e**) Processing workflow for structures obtained from the p97–UBXD1$^{WT}$•ADP dataset. Class 1 corresponds to p97–UBXD1$^{closed}$, class 2 to p97–UBXD1$^{open}$, and class 3 to p97–UBXD1$^{VIM}$. Masks used for the P1 and P6 focused classification and masked local refinement of p97–UBXD1$^{open}$ are shown in transparent blue and yellow, respectively. (**f**) Overlay of all protomers from p97–UBXD1$^{closed}$ (blue) with a protomer in the ADP-bound, down NTD conformation (pink, PDB 5FTK) and a protomer in the ATPγS-bound, up NTD conformation (green, PDB 5FTN), aligned by the D1 large subdomain (residues 211-368). For all protomers, the NTDs are colored, and the D1 and D2 domains are white. (**g**) As in (**f**), but depicting protomers from p97–UBXD1$^{open}$ (blue). (**h**) Nucleotide densities for representative D1 and D2 pockets in p97–UBXD1$^{closed}$ and p97–UBXD1$^{open}$. (**i**) Representative additional density in NTD corresponding to a VIM helix (unsharpened map of P4 in p97–UBXD1$^{open}$).

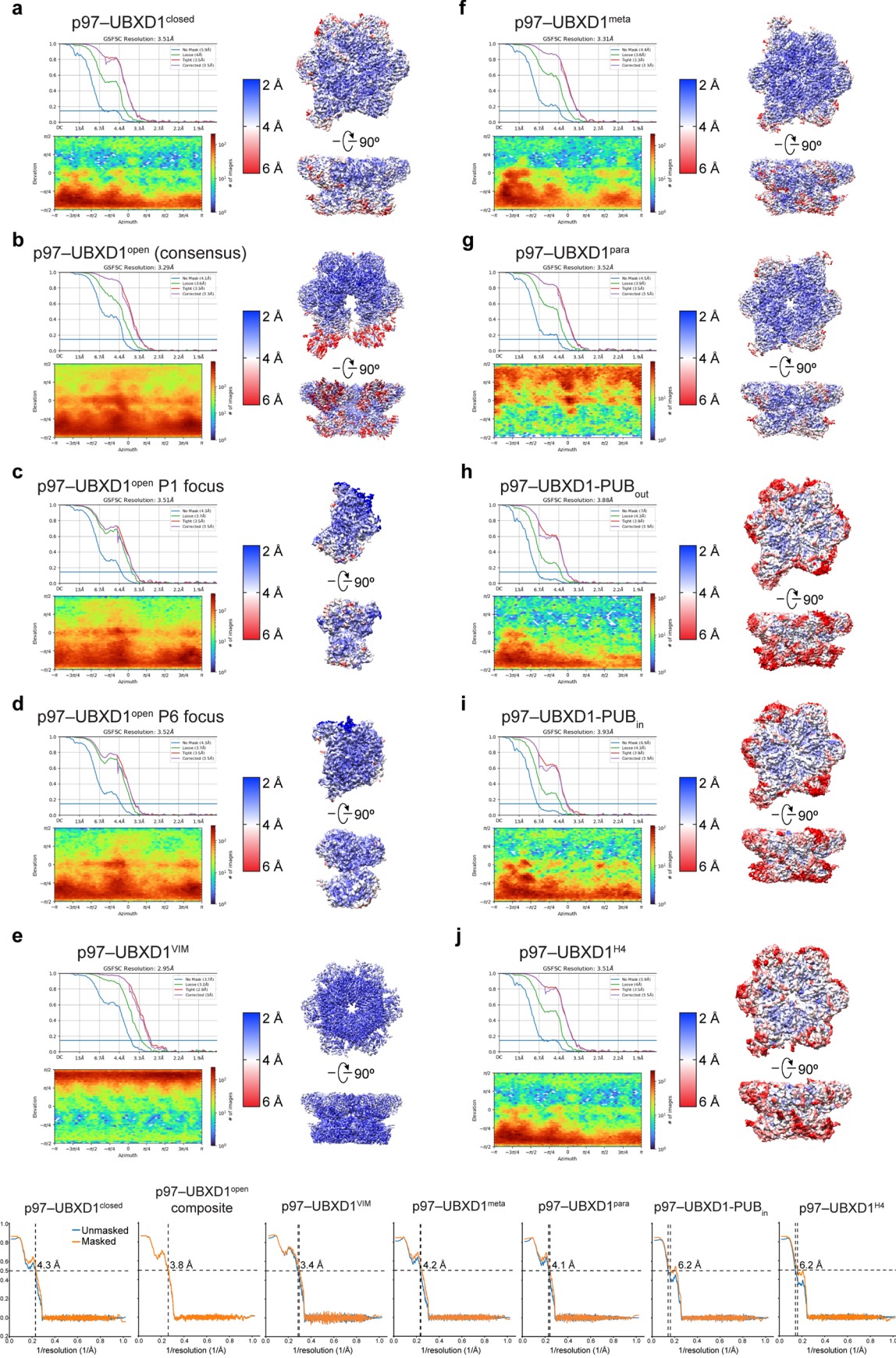

**Extended Data Fig. 2 | See next page for caption.**

**Extended Data Fig. 2 | Cryo-EM densities and resolution estimation from the ADP-bound p97–UBXD1^WT dataset (superstoichiometric). (a** to **j)** Fourier shell correlation (FSC) curves, particle orientation distribution plots, and sharpened density maps colored by local resolution (0.143 cutoff) for (**a**) p97–UBXD1^closed, (**b**) p97–UBXD1^open (consensus map), (**c**) p97–UBXD1^open P1 focus, (**d**) p97– UBXD1^open P6 focus, (**e**) p97–UBXD1^VIM, (**f**) p97–UBXD1^meta, (**g**) p97–UBXD1^para, (**h**) p97–UBXD1-PUB_out, (**i**) p97–UBXD1-PUB_in, and (**j**) p97–UBXD1^H4. (**k**) Map-model FSC curves for p97–UBXD1^closed, p97–UBXD1^open (composite map), p97–UBXD1^VIM, p97–UBXD1^meta, p97–UBXD1^para, p97–UBXD1-PUB_in, and p97–UBXD1^H4. Displayed model resolutions were determined using the masked maps.

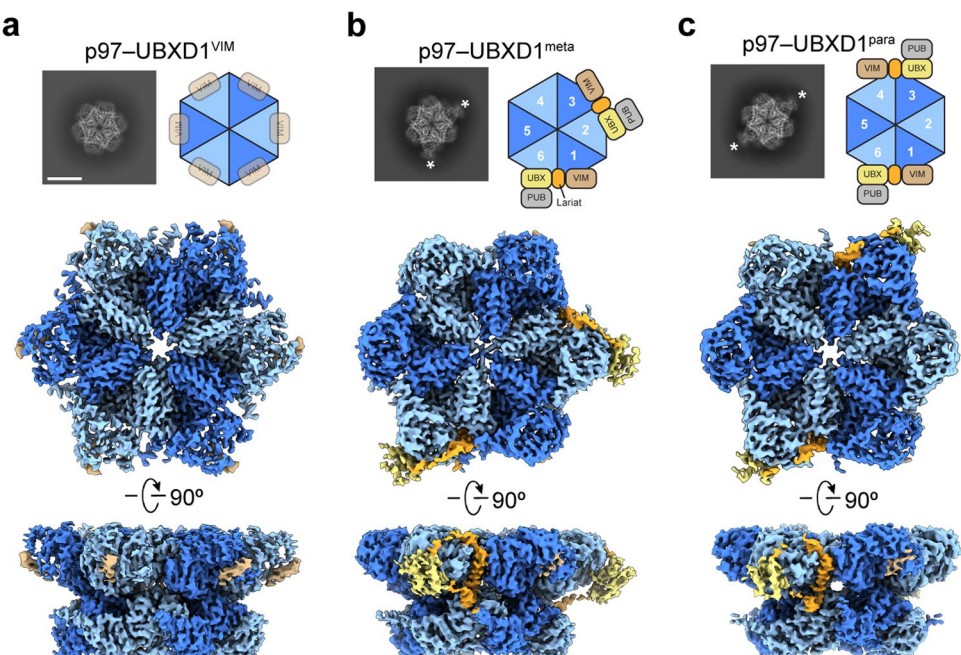

**Extended Data Fig. 3 | Additional configurations of p97–UBXD1 complexes.** (**a to c**) Cartoons, top view projections of sharpened maps showing UBX/PUB density (*), and sharpened maps of (**a**) p97–UBXD1$^{VIM}$, (**b**) p97–UBXD1$^{meta}$, and (**c**) p97–UBXD1$^{para}$ (scale bar equals 100 Å). In p97–UBXD1$^{VIM}$, the VIM density is depicted as a difference map of p97–UBXD1$^{VIM}$ and a map generated from a model without VIM helices.

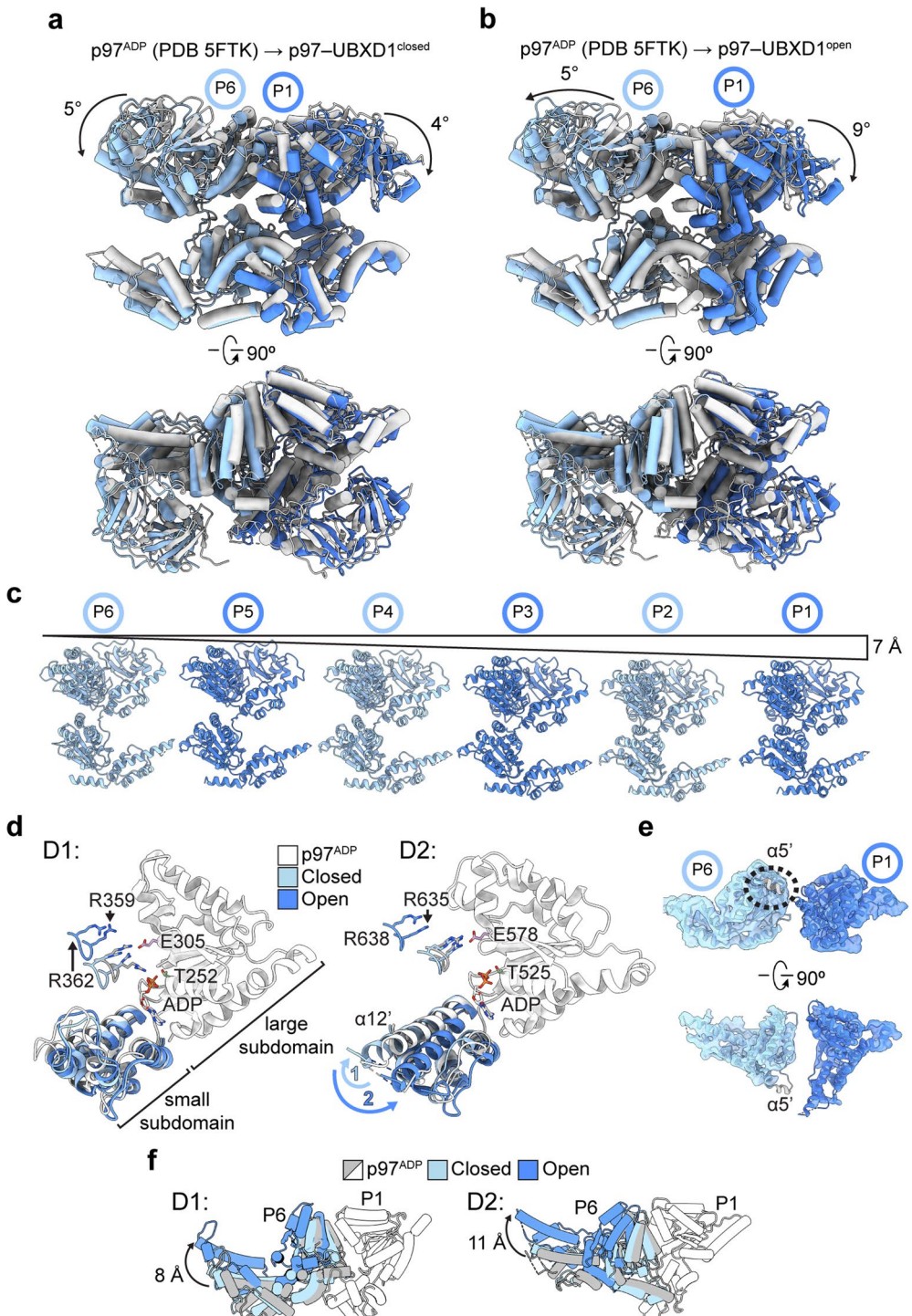

**Extended Data Fig. 4 | Hexamer remodeling in p97–UBXD1^closed and p97– UBXD1^open.** (**a**) Overlay of protomers P1 (dark blue) and P6 (light blue) from p97–UBXD1^closed, aligned to protomers P3 and P4 from PDB 5FTK. P1 and P6 protomers from 5FTK are shown in gray. (**b**) As in (a), but depicting p97– UBXD1^open protomers. (**c**) Side-by-side view of individual protomers aligned based on position in the p97–UBXD1^open hexamer, showing vertical displacement along the pseudo-C6 symmetry axis. (**d**) Overlay of the D1 (left) and D2 (right) AAA+ domains of P1 for p97^ADP, p97–UBXD1^closed, and p97–UBXD1^open, aligned to the large subdomains and colored as indicated. ADP is shown with conserved Walker A/B (green and purple, respectively) and trans-acting (P6) Arg finger

residues indicated. The large rotation of the D2 small subdomain, exemplified by α12', is shown (relative to p97^ADP) for the closed (1) and open (2) states. (**e**) Unsharpened map of the D2 domains of protomers P1 and P6 of p97–UBXD1^open, overlaid with the D2 domain from p97^ADP on P6, showing lack of density for helix α5' (gray, encircled) normally contacting the counterclockwise D2 domain. The D2 domain of P1 of the open state is shown for clarity. (**f**) Top view overlay of the D1 (left) and D2 (right) domains for the P6-P1 pair in the three states and aligned to P1 to show relative rotations of P6, colored as indicated. Rotations shown are from p97^ADP to p97–UBXD1^open and determined from centroid positions of the D1 and D2 domains.

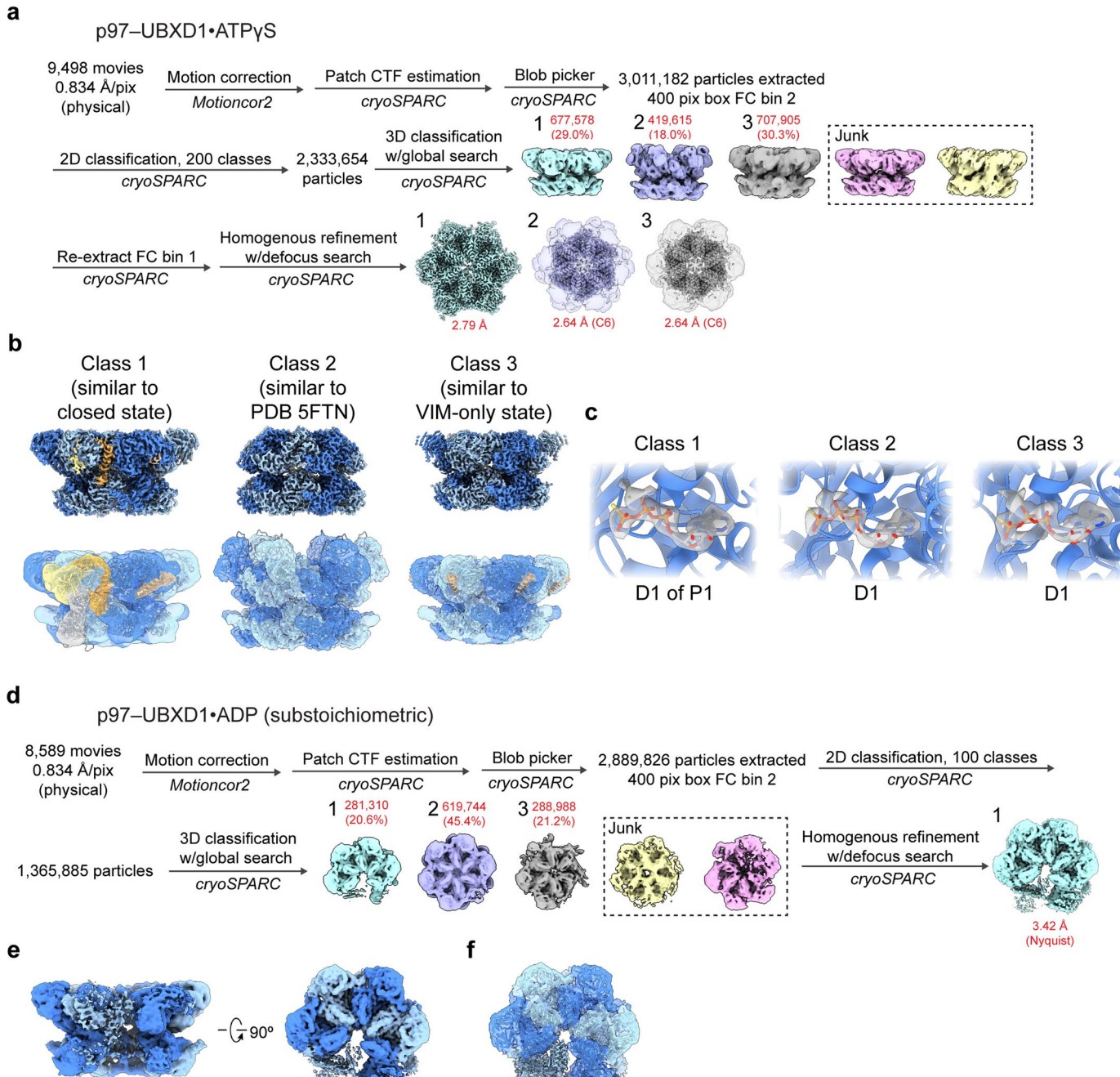

**Extended Data Fig. 5 | Cryo-EM analysis of p97–UBXD1 with ATPγS and with substoichiometric UBXD1.** (**a**) Processing workflow for structures obtained from the p97–UBXD1^WT•ATPγS dataset. Class 1 resembles p97–UBXD1^closed, class 2 is identical to ATPγS-bound p97 (PDB 5FTN), and class 3 resembles p97–UBXD1^VIM. (**b**) (Top row) sharpened maps of class 1-3 refinements. (Bottom row) p97–UBXD1^closed model overlaid with filtered map, unsharpened map, ATPγS-bound p97 hexamer with NTDs in the up state (PDB 5FTN) overlaid with the class 2 unsharpened map, and p97–UBXD1^VIM model overlaid with the class 3 unsharpened map, respectively. All maps are colored as in Fig. 1.

(**c**) Representative nucleotide densities for class 1-3 refinements (sharpened maps), showing clear γ-phosphate and Mg^2+ density. The nucleotide and surrounding binding pocket from PDB 5FTN are shown for clarity. (**d**) Processing workflow for structures from the p97–UBXD1^WT•ADP substoichiometric dataset. Class 1 resembles p97–UBXD1^open, while classes 2 and 3 resemble unbound p97 hexamers in the ADP state. (**e**) Top and side views of class 1 (unsharpened) from (d), colored by p97 protomer. (**f**) Transparent top view of class 1 from (d), overlaid with p97 protomers from the p97–UBXD1^open model.

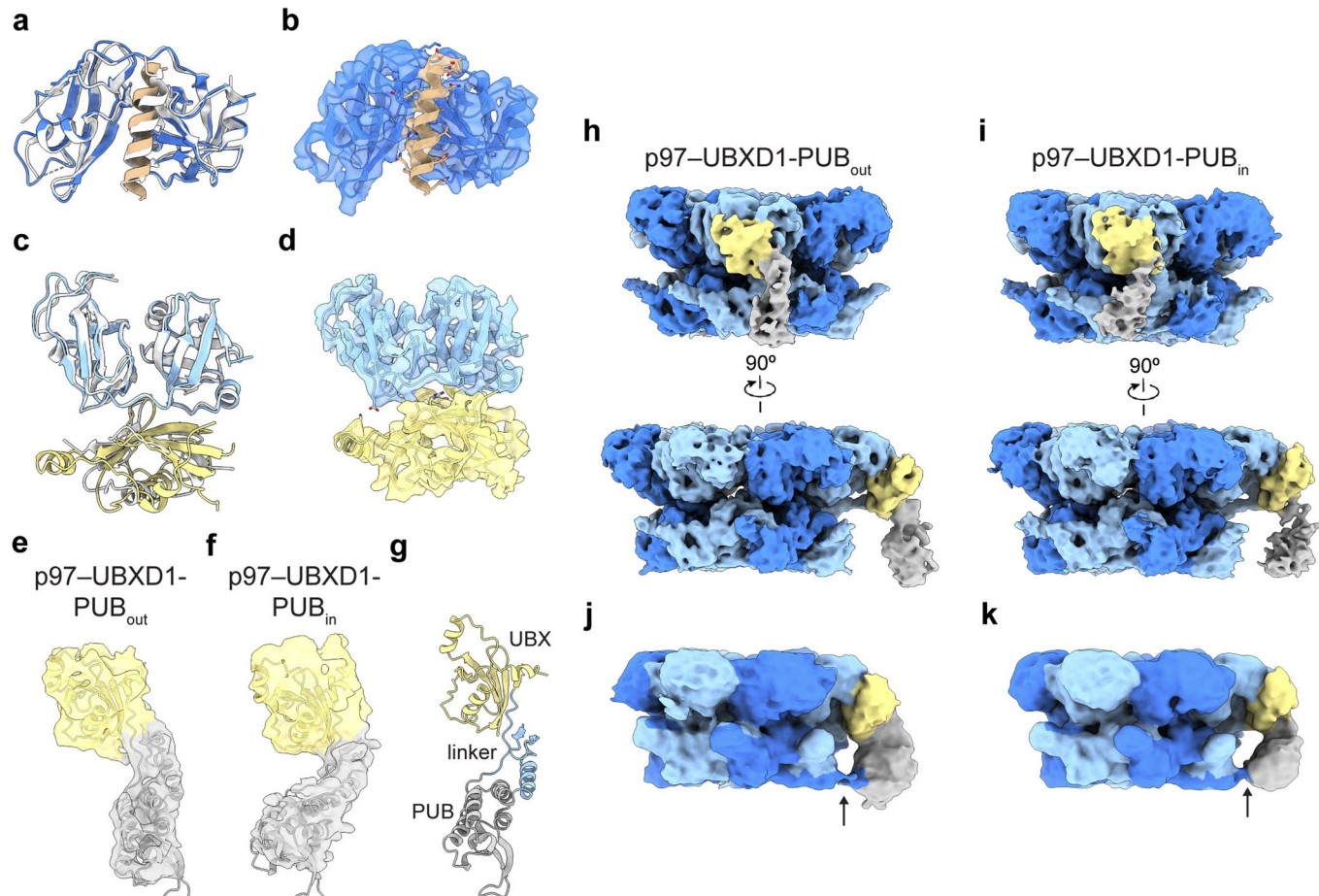

**Extended Data Fig. 6 | VIM, UBX, and PUB structural alignments and validation.** (**a**) Overlay of NTD-VIM from p97–UBXD1closed (colored) and gp78 (PDB 3TIW, white). (**b**) Map and model of the NTD and VIM from p97–UBXD1closed, colored as in (**a**). (**c**) Overlay of NTD-UBX from p97–UBXD1closed (colored) and UBXD7 (PDB 5X4L, white). (**d**) Map and model of the NTD and UBX from p97–UBXD1closed, colored as in (**c**). (**e**) Unsharpened, zoned map and model of UBX and PUB from p97–UBXD1-PUBout. (**f**) Unsharpened, zoned map and model of UBX and PUB from p97–UBXD1-PUBin. (**g**) Model of the UBX (yellow), PUB (gray), and UBX-PUB linker (light blue) from p97–UBXD1closed. (**h**) Unsharpened map of p97–UBXD1-PUBout. The VIM and helical lariat are colored the same as their corresponding protomers for clarity. (**i**) Unsharpened map of p97–UBXD1-PUBin. The VIM and helical lariat are colored the same as their corresponding protomers for clarity. (**j**) Filtered map of p97–UBXD1-PUBout, colored as in (**d**), showing weak density connecting the PUB and P5 CT tail. (**k**) As in (**j**), but for p97–UBXD1-PUBin.

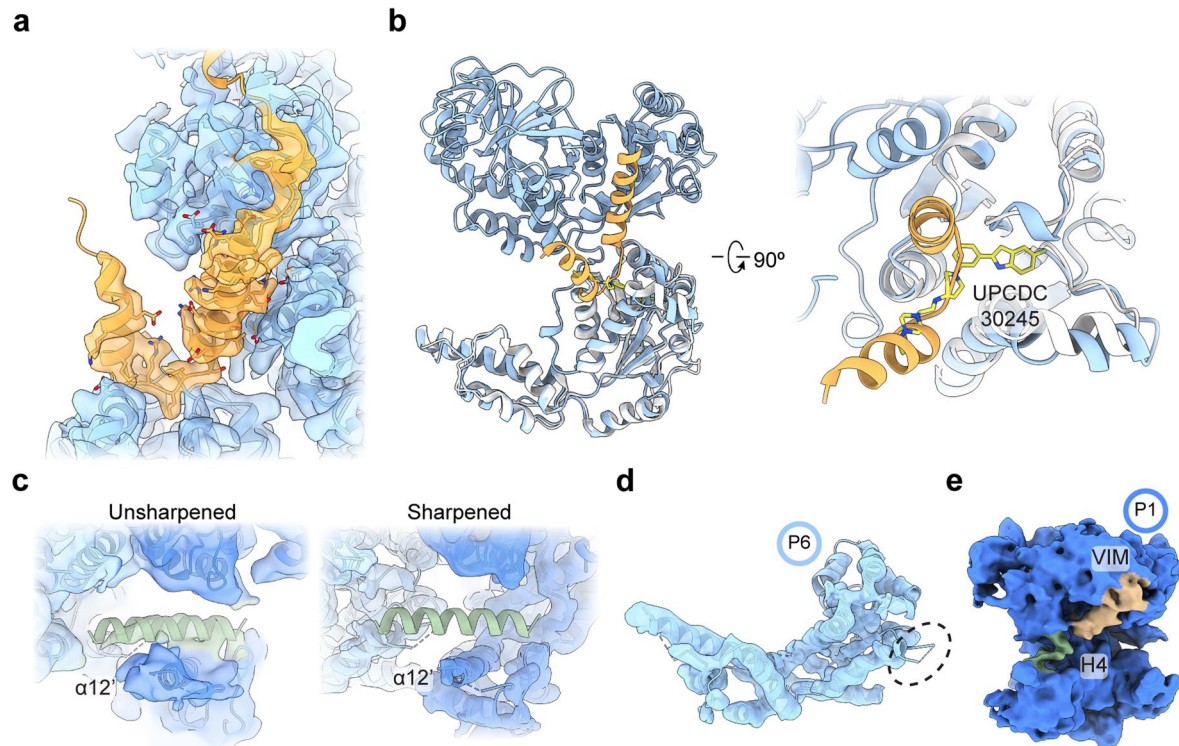

**Extended Data Fig. 7 | Validation of the helical lariat, UPCDC30245 binding, and additional structural features of p97–UBXD1^H4 and p97–UBXD1^open.** (**a**) Sharpened map and model of Lα2-4, connecting strands of the helical lariat, and adjacent regions of P6 of p97–UBXD1^meta, colored as in Fig. 1. (**b**) Overlay of P6 (blue) and Lα3-4 (orange) from p97–UBXD1^closed with a p97 protomer (white) bound to UPCDC30245 (yellow) (PDB 5FTJ), aligned by the D2 domain (residues 483-763). (**c**) View of H4 and surrounding p97 density in p97–UBXD1^H4 (left: unsharpened, right: sharpened) overlaid with the model for this state. (**d**) Sharpened map and model of the D2 domain of protomer P6 of p97–UBXD1^H4, showing lack of density for α5′ (encircled). (**e**) Unsharpened map of protomer P1 of p97–UBXD1^open. Density putatively corresponding to H4 is colored in green.

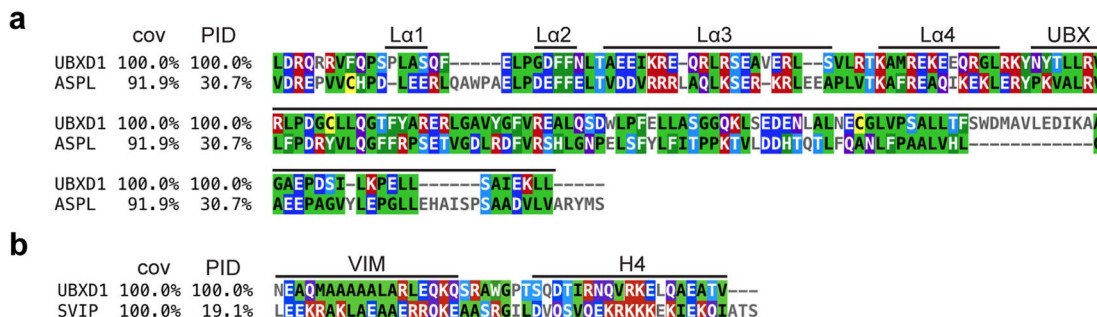

**Extended Data Fig. 8 | Conservation of UBXD1, ASPL, and SVIP sequences.**
(**a**) Alignment of UBX-lariat sequences from UBXD1 (residues 270-441) and ASPL (residues 318-495). Structural elements in the UBXD1 sequence are indicated above. Cov = covariance relative to the human sequence, Pid = percent identity relative to the human sequence. (**b**) Alignment of VIM–H4 sequences from UBXD1 (residues 50-93) and SVIP (residues 18-64). Structural elements in the UBXD1 sequence are indicated above.

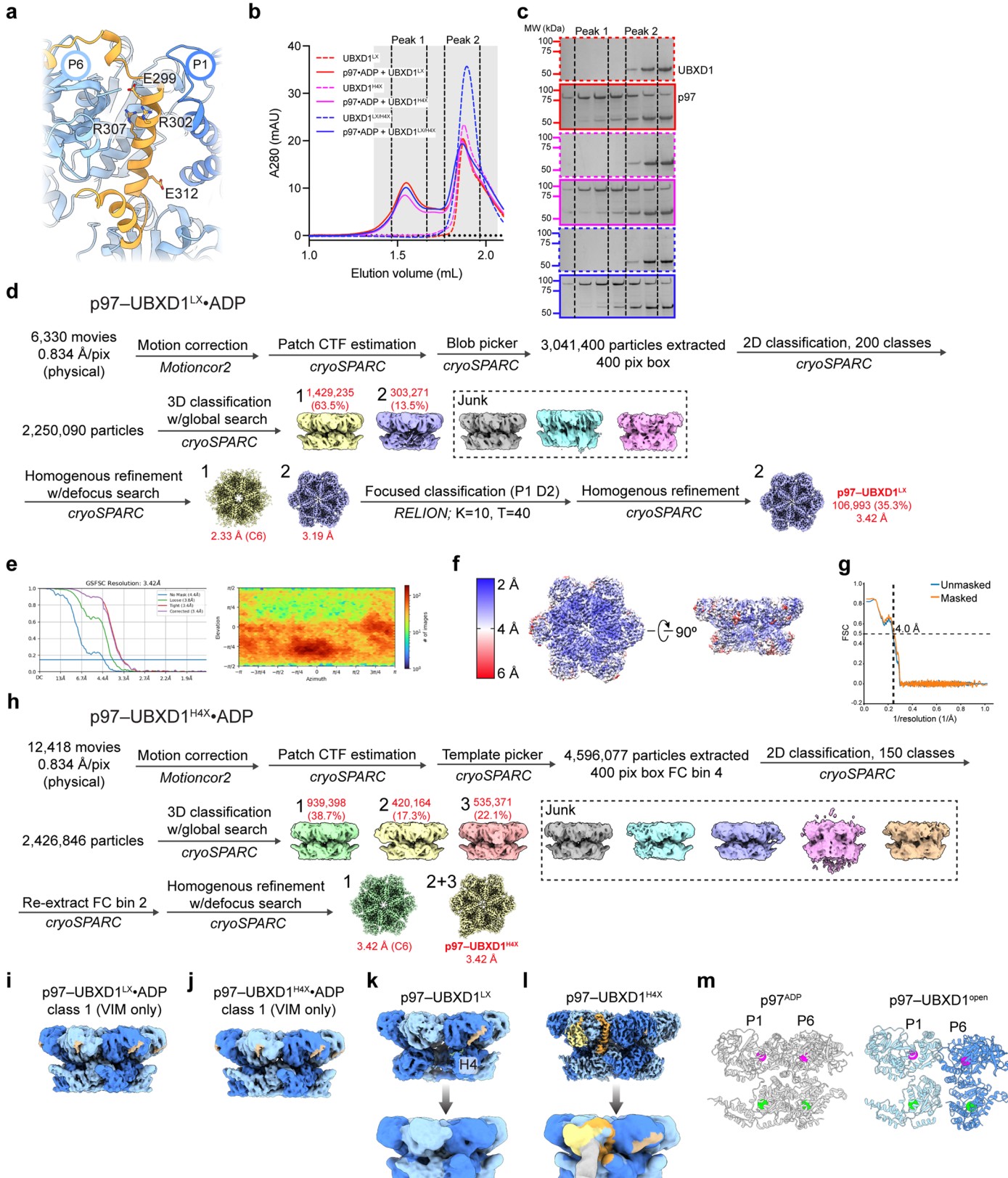

**Extended Data Fig. 9 | See next page for caption.**

**Extended Data Fig. 9 | Biochemical and cryo-EM analysis of lariat and H4 mutations.** (**a**) Residues mutated in Lα3 of the UBXD1 helical lariat, shown on p97–UBXD1^closed^. (**b**) SEC traces of UBXD1 mutants alone or incubated with p97 and ADP, showing a left shift in peak elution volume for p97 samples with UBXD1. Fractions in the shaded range were analyzed by SDS–PAGE. Data are representative of three independently performed experiments with similar results. (**c**) Coomassie Brilliant Blue-stained SDS–PAGE gels of fractions from SEC runs in (**b**). Data are representative of three independently performed experiments with similar results. (**d**) Cryo-EM processing workflow for the p97–UBXD1^LX^•ADP dataset. Class 1 is identical to p97–UBXD1^VIM^; class 2 has VIM and H4 density. (**e**) FSC curve and particle orientation distribution plot for p97–UBXD1^LX^. (**f**) Sharpened density map colored by local resolution (0.143 cutoff) of p97–UBXD1^LX^. (**g**) Map-model FSC for p97–UBXD1^LX^. Displayed resolution was determined using the masked map. (**h**) Cryo-EM processing workflow for the p97–UBXD1^H4X^•ADP dataset. Class 1 resembles p97–UBXD1^VIM^; classes 2 and 3 resemble p97–UBXD1^closed^. (**i**) Unsharpened map of class 1 from p97–UBXD1^LX^•ADP dataset. (**j**) Unsharpened map of class 1 from p97–UBXD1^H4X^•ADP. (**k**) Top: unsharpened map of p97–UBXD1^LX^. Note lack of density for the D2 domain of the protomer clockwise from the VIM-H4 bound protomer. Bottom: filtered map, colored only by p97 density, confirming that density for the aforementioned D2 domain is present. (**l**) Top: sharpened map of p97–UBXD1^H4X^, colored as in Fig. 1. Bottom: filtered map, confirming that density for the PUB domain is present. (**m**) Positions of calculated centroids for D1 (residues 210-458, magenta spheres) and D2 domains (residues 483-763, green spheres) of P1 and P6, for p97^ADP^ and p97–UBXD1^open^.

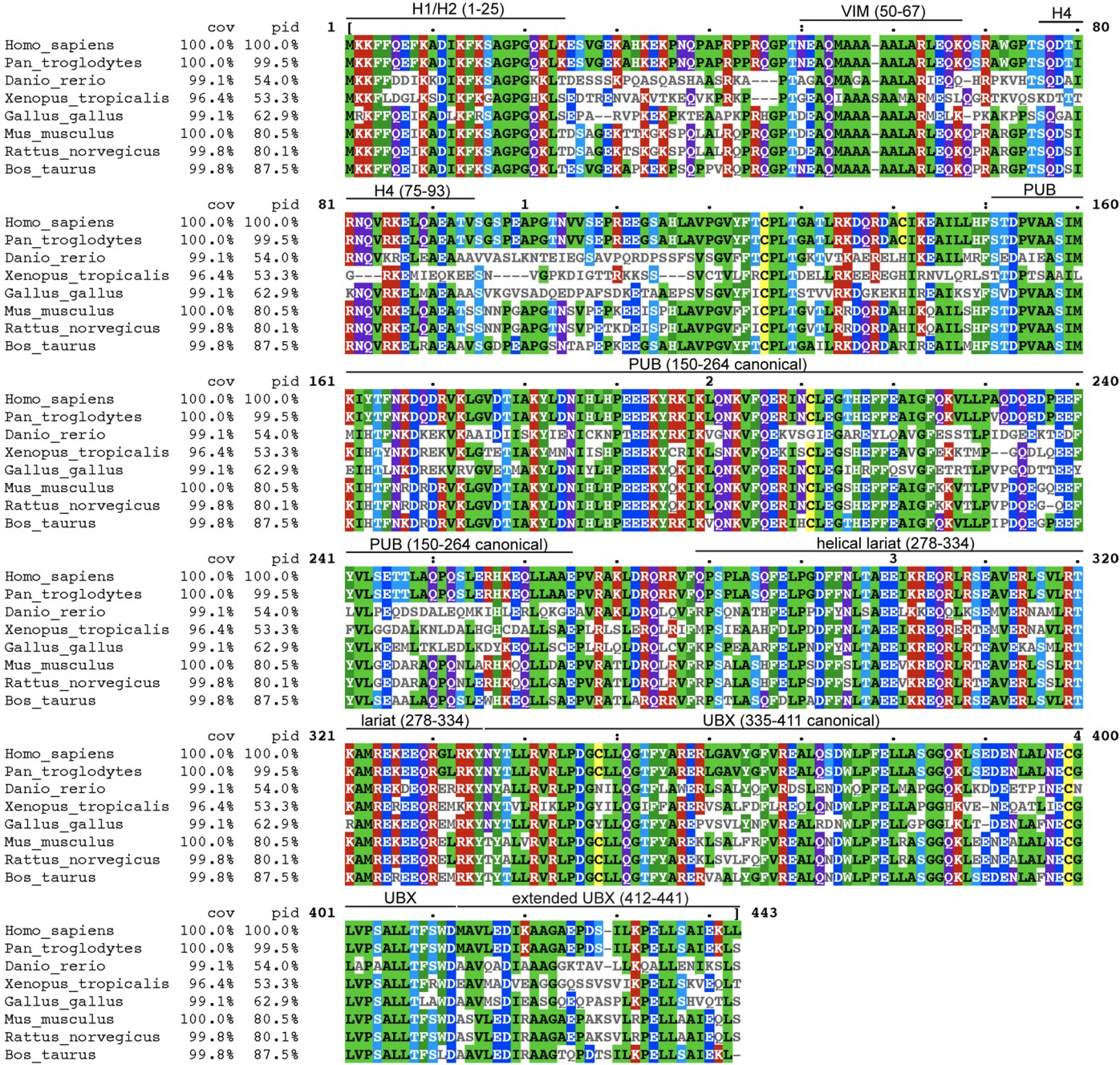

**Extended Data Fig. 10 | Alignment of UBXD1 sequences.** Multiple sequence alignment of UBXD1 homologs from *Homo sapiens*, *Mus musculus*, *Xenopus tropicalis*, *Gallus gallus*, *Rattus norvegicus*, *Bos taurus*, *Pan troglodytes*, and *Danio rerio*. Structural elements and residue ranges (in the human sequence) are marked above. Cov = covariance relative to the human sequence, Pid = percent identity relative to the human sequence.

# Reporting Summary

## Statistics

For all statistical analyses, confirm that the following items are present in the figure legend, table legend, main text, or Methods section.

| n/a | Confirmed | |
|---|---|---|
| ☐ | ☒ | The exact sample size (*n*) for each experimental group/condition, given as a discrete number and unit of measurement |
| ☐ | ☒ | A statement on whether measurements were taken from distinct samples or whether the same sample was measured repeatedly |
| ☒ | ☐ | The statistical test(s) used AND whether they are one- or two-sided<br>*Only common tests should be described solely by name; describe more complex techniques in the Methods section.* |
| ☒ | ☐ | A description of all covariates tested |
| ☒ | ☐ | A description of any assumptions or corrections, such as tests of normality and adjustment for multiple comparisons |
| ☐ | ☒ | A full description of the statistical parameters including central tendency (e.g. means) or other basic estimates (e.g. regression coefficient) AND variation (e.g. standard deviation) or associated estimates of uncertainty (e.g. confidence intervals) |
| ☒ | ☐ | For null hypothesis testing, the test statistic (e.g. *F*, *t*, *r*) with confidence intervals, effect sizes, degrees of freedom and *P* value noted<br>*Give P values as exact values whenever suitable.* |
| ☒ | ☐ | For Bayesian analysis, information on the choice of priors and Markov chain Monte Carlo settings |
| ☒ | ☐ | For hierarchical and complex designs, identification of the appropriate level for tests and full reporting of outcomes |
| ☒ | ☐ | Estimates of effect sizes (e.g. Cohen's *d*, Pearson's *r*), indicating how they were calculated |

*Our web collection on statistics for biologists contains articles on many of the points above.*

## Software and code

Policy information about availability of computer code

| Data collection | Electron microscopy data were collected using SerialEM 4.1.0, chromatography data were collected using UNICORN 7, and ATPase data were collected using SoftMax Pro v7. |
|---|---|
| Data analysis | Data were analyzed using MotionCor2 v1.6.4, cryoSPARC v3.3, RELION v3.1, Coot v0.8.9.2, ISOLDE v1.3, Phenix v1.20.1, Rosetta v3.12, AlphaFold2, UCSF Chimera v1.16, UCSF ChimeraX v1.3, GraphPad Prism v9.3.1, MUSCLE v5, MView v1.67, DALI v5, and Peptide Nexus Sequence Scrambler (https://peptidenexus.com/article/sequence-scrambler). |

For manuscripts utilizing custom algorithms or software that are central to the research but not yet described in published literature, software must be made available to editors and reviewers. We strongly encourage code deposition in a community repository (e.g. GitHub). See the Nature Portfolio guidelines for submitting code & software for further information.

## Data

Policy information about availability of data

All manuscripts must include a data availability statement. This statement should provide the following information, where applicable:
- Accession codes, unique identifiers, or web links for publicly available datasets
- A description of any restrictions on data availability
- For clinical datasets or third party data, please ensure that the statement adheres to our policy

Cryo-EM densities have been deposited at the Electron Microscopy Data Bank under accession codes EMD: 28982 (p97:UBXD1 closed), EMD: 28983 (p97:UBXD1

open composite), EMD: 28984 (p97:UBXD1 open consensus), EMD: 28985 (p97:UBXD1 open P1 focused map), EMD: 28986 (p97:UBXD1 open P6 focused map), EMD: 28987 (p97:UBXD1 VIM), EMD: 28988 (p97:UBXD1 meta), EMD: 28989 (p97:UBXD1 para), EMD: 28990 (p97:UBXD1-PUBin), EMD: 28991 (p97:UBXD1 H4), and EMD: 28992 (p97:UBXD1 LX). Atomic coordinates have been deposited at the Protein Data Bank under accession codes PDB: 8FCL (p97:UBXD1 closed), PDB: 8FCM (p97:UBXD1 open), PDB: 8FCN (p97:UBXD1 VIM), PDB: 8FCO (p97:UBXD1 meta), PDB: 8FCP (p97:UBXD1 para), PDB: 8FCQ (p97:UBXD1-PUBin), PDB: 8FCR (p97:UBXD1 H4), and PDB: 8FCT (p97:UBXD1 LX). Accession codes for additional models referenced in this study are: PDB: 5FTK (p97:ADP), PDB: 5FTN (p97:ATPγS), PDB: 5FTJ (p97:UPCDC30245) PDB: 5IFS (p97:ASPL-C), PDB: 3TIW (NTD:gp78-VIM), PDB: 5X4L (NTD:UBXD7-UBX), AF-Q9BZV1-F1 (UBXD1 AlphaFold model), and AF-Q8NHG7-F1 (SVIP AlphaFold model). Uncropped images for Extended Data Figs. 1b and 7e and data used for all biochemical experiments are provided as Source Data online.

## Human research participants

Policy information about <u>studies involving human research participants and Sex and Gender in Research.</u>

| | |
|---|---|
| Reporting on sex and gender | N/A |
| Population characteristics | N/A |
| Recruitment | N/A |
| Ethics oversight | N/A |

Note that full information on the approval of the study protocol must also be provided in the manuscript.

# Field-specific reporting

Please select the one below that is the best fit for your research. If you are not sure, read the appropriate sections before making your selection.

☒ Life sciences ☐ Behavioural & social sciences ☐ Ecological, evolutionary & environmental sciences

For a reference copy of the document with all sections, see <u>nature.com/documents/nr-reporting-summary-flat.pdf</u>

# Life sciences study design

All studies must disclose on these points even when the disclosure is negative.

| | |
|---|---|
| Sample size | Cryo-EM images were collected for each sample until there was a reasonable expectation of performing reconstructions at the desired resolution (~3-4 Angstrom, sufficient for atomic model building). In all cases the desired resolutions were reached, indicating that the sample sizes are sufficient. Biochemical experiments were performed in biological triplicate and technical triplicate (where applicable) in keeping with standard practice in the field. |
| Data exclusions | Micrographs were excluded based on poor maximum estimated resolution. Particles were excluded during 2D and 3D classification in cryoSPARC and RELION based on assignment to low-resolution or otherwise artifactual classes. |
| Replication | Cryo-EM data were randomly divided into two halves that were independently refined to achieve the reported resolutions. ATPase assays were performed in biological and technical triplicate. Chromatography and SDS-PAGE experiments were performed in biological triplicate. All replication attempts were successful. |
| Randomization | Cryo-EM data was randomly assigned to two half sets during refinement. Resolution estimates are based on comparisons of reconstructions from these half sets. Other experiments were not randomized as no subjective assessment of data was required. |
| Blinding | Investigators were not blinded in any cryo-EM or biochemical data collection or analysis as no subjective assessment of data was required. |

# Reporting for specific materials, systems and methods

We require information from authors about some types of materials, experimental systems and methods used in many studies. Here, indicate whether each material, system or method listed is relevant to your study. If you are not sure if a list item applies to your research, read the appropriate section before selecting a response.

## Materials & experimental systems

| n/a | Involved in the study |
|-----|----------------------|
| ☒ | Antibodies |
| ☐ | ☒ Eukaryotic cell lines |
| ☒ | Palaeontology and archaeology |
| ☒ | Animals and other organisms |
| ☒ | Clinical data |
| ☒ | Dual use research of concern |

## Methods

| n/a | Involved in the study |
|-----|----------------------|
| ☒ | ChIP-seq |
| ☒ | Flow cytometry |
| ☒ | MRI-based neuroimaging |

# Eukaryotic cell lines

Policy information about cell lines and Sex and Gender in Research

| | |
|---|---|
| Cell line source(s) | Sf9 insect cells were obtained from Expression Systems. |
| Authentication | The cell line used was not authenticated. |
| Mycoplasma contamination | The cell line used was not tested for Mycoplasma contamination. |
| Commonly misidentified lines (See ICLAC register) | No commonly misidentified cell line was used in this study. |

