## [Peer Review File · Nature Structural & Molecular Biology]

Peer Review Information

Manuscript Title: The p97/VCP adapter UBXD1 drives AAA+ remodeling and ring opening through multi-domain tethered interactions

Corresponding author name(s): Daniel Southworth, Michelle Arkin

Reviewer Comments & Decisions:

Decision Letter, initial version:

Message: 17th May 2023

Dear Dr Southworth,

Thank you again for submitting your manuscript "The p97/VCP adapter UBXD1 drives AAA+ remodeling and ring opening through multi-domain tethered interactions". I apologize for the delay in responding, which resulted from the difficulty in obtaining suitable referee reports. Nevertheless, we now have comments (below) from the 2 reviewers who evaluated your paper. In light of those reports, we remain interested in your study and would like to see your response to the comments of the referees, in the form of a revised manuscript.

You will see that while reviewers appreciate the results, they raise concerns which should be addressed in a revision. Specifically, reviewer #1 questions the biological significance of the findings, which we would ask you to investigate further. In line with reviewer #2 suggestions, we would encourage you to further expand the mutagenesis experiments, to strengthen the manuscript.

Please be sure to address/respond to all concerns of the referees in full in a point-by-point response and highlight all changes in the revised manuscript text file.

We appreciate the requested revisions are extensive. We thus expect to see your revised manuscript within 6 months. If you cannot send it within this time, please let us know. We will be happy to consider your revision as long as nothing similar has been accepted for

publication at NSMB or published elsewhere. Should your manuscript be substantially delayed without notifying us in advance and your article is eventually published, the received date would be that of the revised, not the original, version.

Reporting Summary:

When submitting the revised version of your manuscript, please pay close attention to our [href="https://www.nature.com/nature-portfolio/editorial-policies/image-integrity">Digital Image Integrity Guidelines. and to the following points below:](https://www.nature.com/nature-portfolio/editorial-policies/image-integrity)

Please note that all key data shown in the main figures as cropped gels or blots should be presented in uncropped form, with molecular weight markers. These data can be aggregated into a single supplementary figure. While these data can be displayed in a relatively informal style, they must refer back to the relevant figures. These data should be submitted with the last revision, prior to acceptance, but you may want to start putting it together at this point.

SOURCE DATA: we request that authors provide, in tabular form, all the data underlying the graphical representations used in figures. This is to further increase transparency in data reporting, as detailed in this editorial (<http://www.nature.com/nsmb/journal/v22/n10/full/nsmb.3110.html>). Spreadsheets can be submitted in excel format. Only one (1) file per figure is permitted; thus, for multi-paneled figures, the source data for each panel should be clearly labeled in the Excel file; alternately the data can be provided as multiple, clearly labeled sheets in an Excel file. When submitting files, the title field should indicate which figure the source data pertains to. We request our authors to provide source data at the revision stage, so that they are part of the

peer-review process. Please also include the uncropped blots and gels in the Source data file.

We require deposition of coordinates (and, in the case of crystal structures, structure factors) into the Protein Data Bank with the designation of immediate release upon publication (HPUB). Electron microscopy-derived density maps and coordinate data must be deposited in EMDB and released upon publication. Deposition and immediate release of NMR chemical shift assignments are highly encouraged. Deposition of deep sequencing and microarray data is mandatory, and the datasets must be released prior to or upon publication. To avoid delays in publication, dataset accession numbers must be supplied with the final accepted manuscript and appropriate release dates must be indicated at the galley proof stage. Please find the complete NRG policies on data availability at <http://www.nature.com/authors/policies/availability.html>.

[redacted]

Sincerely,

Katarzyna Ciazynska
(she/her)
Associate Editor
Nature Structural & Molecular Biology
<https://orcid.org/0000-0002-9899-2428>

Referee expertise:

Referee #1: cryo-EM, protein homeostasis

Referee #2: cryo-EM, protein degradation

Reviewers' Comments:

Reviewer #1:

Remarks to the Author:

The manuscript by Braxton et al. reports the structure of the p97 AAA+ ATPase in complex with the poorly characterized UBXD1 adapter. p97 is an essential molecular machine that is known to unfold proteins, and its interactions with more than thirty known adapters drive its functions in many cellular pathways. Many structures of p97 have been determined in recent years, including those in active and inactive conformations and bound to various adapters. These structures have enhanced our understanding of how p97 unfolds substrate, but there are still notable gaps in knowledge of how it is regulated by so many adapters.

The major focus of the study describes how p97 is rendered in an inactive state by UBXD1 through a network of interactions that partially split the hexamer. This mode of p97 inhibition is novel, though the structurally related ASPL inactivates p97 by complete disassembly of hexamers (Arumughan et al., Nat. Commun. 2016). The present manuscript therefore establishes an alternative mode of p97 inactivation, which broadens understanding of the diversity of p97 regulatory mechanisms.

The manuscript is well written and the structural work is excellent, but enthusiasm of the manuscript is dampened by the lack of insight into the biological significance of the structure. The open structure was derived from a small percentage of particles after multiple rounds of classification, which makes it unclear if the structure represents a biologically relevant state of the complex. Some interesting ideas are proposed in the Discussion about the potential role of the open state during substrate translocation or release, but there is no indication that the complex interacts with substrate. Overall, the work is of high technical quality and the partial splitting of p97 by UBXD1 is interesting, but the impact of the manuscript does not rise to the level of NSMB.

The manuscript would be strengthened by addressing how the structure fits in the context of regulating p97 function in UBXD1-associated pathways. Previous studies indicate a relationship between UBXD1 and a variety of cellular functions, including turnover of damaged organelles, ERAD, endolysosomal sorting, and so forth. It would be fascinating to know the role of the UBXD1 in the functional cycle of p97 in such pathways.

Reviewer #2:

Remarks to the Author:

This manuscript is an impressive cryo-EM study of how UBXD1 interacts with and inhibits the p97 ATPase. A multivalent interaction mechanism is uncovered, with a state that causes the p97 hexamer to split apart at one of the inter-subunit seams. New structural motifs are discovered including a lariat structure that wraps around a p97 N-terminal domain with two helices interacting at the protomer interface. The UBXD1 PUB domain,

which interacts with the p97 C-terminal HbYX motif, is also observed below the UBX domain. Two conformational states are reported for the PUB domain that differ by a 46° rotation. The multidomain interactions of UBXD1 enable it to span three p97 subunits. One aspect of the study that is a little more challenging to understand is the observation of a VIM-only complex, and as mentioned below some more information could be given regarding this state. In addition, it is somewhat surprising that the LX/H4X mutations didn't have a bigger effect in Figure 5e. Inclusion of experimental data that the LX and H4X mutations led to loss of binding for these regions might strengthen this part of the manuscript albeit the loss of density in the cryoEM structures does provide some experimental support for the efficacy of these mutations. Overall, this manuscript is a sizable contribution to the p97 field by providing atomic level resolution structures of its complex with UBXD1 and thereby revealing the impact of UBXD1 on p97 structure and activity. It is of immediate interest to people in the p97 and protein quality control fields and there are no flaws that should prevent this manuscript's publication; however, some suggestions are made below.

p. 2, line 60; 'subtracts' should be 'substrates'.

Extended Data Fig. 1b: molecular weight markers should be included on the lower image panels.

p. 4, line 118-119: this Reviewer can't see the stated 'modest shift in elution' for ADP compared to ATPgammaS. Either clarification is needed or perhaps this statement should be deleted. This also appears on p. 7, line 193-194.

Fig. 1e,f: the two previously uncharacterized helices (in orange?) and N-terminal linker (not clear but probably also in orange?) should be labeled as it's not easy to visualize them in this figure.

p. 6, line 180: it's not clear how the authors conclude the additional helical density in the NTD cleft of all protomers is from UBXD1 VIM. Is the resolution for this helix sufficient that this density could be readily mapped as VIM? What dictates whether VIM displaces the UBX and other contacts? Mechanistically, how does it prevent UBX binding; does it have higher affinity and bind to an overlapping site for example?

p. 7, line 211: 'seem' should probably be 'seam'

Author Rebuttal to Initial comments

Response to Reviewers' comments

We thank the Reviewers for their time and thoughtful consideration of this manuscript. Both Reviewers commented positively about the work, indicating "*The manuscript is well written and the structural work is excellent*" (R1), and "*this manuscript is a sizable contribution to the p97 field by providing atomic level resolution structures of its complex with UBXD1 and thereby revealing the impact of UBXD1 on p97 structure and activity*" (R2). Additionally, the Reviewers each identify a number of important concerns and clarifications which we have specifically addressed below.

General concerns raised about the manuscript include the biological significance of the determined structures, the modest effects of helical lariat/H4 mutations on UBXD1 inhibition of p97 ATPase activity, and the significance of VIM-only states in our cryo-EM data. Based on these concerns we have determined additional structures of p97:UBXD1 with a substoichiometric amount of UBXD1, which likely mimics the relative concentrations of these proteins in cells (Fig. 2c, Extended Data Fig. 4d-f). Notably, of the particles with UBXD1 bound, 100% are in the open state, supporting the assignment of this structure as functionally relevant. We also validated that the helical density found in many p97 NTDs indeed corresponds to the UBXD1 VIM by mutating this helix and collecting additional cryo-EM data (Fig. 1 for Reviewers). This analysis revealed that no helical density is present in the NTD cleft, only the globular density of the UBXD1 UB domain, and confirms that the observed helical density in samples with wild-type UBXD1 is indeed the VIM. To further emphasize the biological significance of our study, we have (in the Discussion) more directly connected our structures with previously published data indicating that UBXD1 may have a role in recruiting non-ubiquitylated substrates to p97 using the H1/H2 domain, in addition to work demonstrating that UBXD1 cooperates with ubiquitin-binding adapters in specific p97-dependent pathways. Based on Reviewer 2's important question about the modest effect of UBXD1 mutants on p97 ATPase activity, we have tested two additional mutants in this assay: an N-terminal construct containing only the H1/H2, VIM, and H4 domains, a C-terminal construct containing only the PUB, lariat, and UB domains (Fig. 6a). Importantly, both of these constructs are severely impaired in p97 ATPase inhibition, indicating that tethering of a single UBXD1 molecule across multiple protomers is essential for this inhibitory function (Fig. 6c,d). Furthermore, combining both UBXD1 halves did not restore activity, supporting this assertion. These results further suggest that the p97:UBXD1 interaction is highly multivalent, such that mutating one UBXD1 domain does not strongly impact activity but removing several at once does.

Together these additional experiments and analyses further support our conclusions and provide a more comprehensive view of the p97:UBXD1 interaction. Of note, during revision of this manuscript a study from the groups of Hemmo Meyer and Christine Beuck was published in *Nature Communications* that identifies a previously unknown ubiquitin-binding activity of the UBXD1 UB domain, supporting a direct role for UBXD1 in p97 substrate processing and elevating the biological significance of the architecture we define in our study¹. However, similar to other studies but in contrast to ours, they conclude that the UBXD1 UB domain does not interact with NTD but instead interacts distally, adjacent to the D2 ring. Thus, our study remains the only definitive structural characterization of this enigmatic adapter, clarifying a number of previous proposals about its architecture and interactions with p97 and revealing an entirely novel mechanism for adapter control of p97 function through its multi-tethered interactions.

Responses to specific comments and concerns:

Reviewer 1:

“...enthusiasm of the manuscript is dampened by the lack of insight into the biological significance of the structure. The open structure was derived from a small percentage of particles after multiple rounds of classification, which makes it unclear if the structure represents a biologically relevant state of the complex.” Reviewer 1 brings up an important point that our open structure was derived from a relatively small population of data following classification and raises concerns that this impacts the biological significance. Performing multiple rounds of classification is a standard and essential procedure for nearly all modern high-resolution cryo-EM structure determination methods. These methods are necessary to remove particle artifacts selected by the imperfect procedures used by automated particle picking (‘junk’ data, including ice contamination, protein aggregates, or denatured particles that are common due to the vitrification process) and to classify different particle conformations or compositions. The main criterion of this selection process is to obtain datasets that lead to the highest resolution final 3D maps. Thus, it is common for the final particle number to be relatively small compared to the initial data collected.

We agree that in some cases, smaller percentages of particles in a particular class may lead to questions about the biological significance relative to other classes. However, we strongly disagree that this is the case here for either the open or closed states of p97:UBXD1 because of the structural arrangement of UBXD1. Notably, in these two states UBXD1 is making considerable contact with p97, interacting via every major domain across multiple p97 protomers. We now calculate that in the open and closed states, UBXD1 interactions bury ~3200 Å² each. This is substantial and in the range for large interfaces and protein:protein interactions that induce large conformational changes, and thus is highly supportive of a biologically significant interaction². This is in strong contrast to the class that contains the highest percentage of particles, the p97:UBXD1^{VIM} state, in which only the VIM helix of UBXD1 is bound to p97 (VIM-only state). This interaction buries only ~700 Å² and induces no conformational changes in the p97 hexamer. We have added these points to the Results, lines: 172-174, 184-185.

We postulate that the ‘VIM-only’ state contains the predominant percentage of particles because of our *in vitro* incubation conditions in which UBXD1 is incubated at high, super-stoichiometric concentrations (20 μM compared to 10 μM monomer for p97). This was performed in order to maximize the number of p97 hexamers bound to UBXD1. However, these conditions likely favor VIM-binding to every available p97 NTD, thereby outcompeting UBX-lariat interactions that comprise the asymmetric open and closed states. In response to the Reviewer’s concerns we have now performed an additional cryo-EM data collection and analysis in which UBXD1 is incubated at 1.7 μM (with 10 μM p97), which reflects a 1 UBXD1: 6 p97 monomer ratio. These relative concentrations likely mimic those found in cells, as p97 is one of the most abundant proteins in eukaryotes, and present in excess of most if not all adapters³. Under these conditions ~75% of p97 hexamers do not have any UBXD1 density, as expected given the lower concentration used in this experiment. Strikingly, however, the remaining 25% of particles are bound to UBXD1 and exclusively in the open state in which the UBX-lariat and VIM domains interact with the protomers at the open interface. Thus, under sub-stoichiometric conditions,

100% of the UBXD1-bound p97 particles adopt the open state, strongly supporting the biological significance of this conformation relative to the other conformations observed. We have added these data to Fig. 2c and Extended Data Fig. 4d-f and included a discussion of the results in lines: 248-268). We thank the Reviewer for bringing up this point.

We also note that we only observe the p97:UBXD1 open state in the presence of ADP as the nucleotide. When p97 is incubated with ATPγS, we are able to resolve the UBXD1-bound closed state, but not the open state, indicating a post-hydrolysis ADP state may be necessary for the open conformation. This makes sense given that the nucleotide pockets reside at the protomer interface and hydrolysis destabilizes interprotomer nucleotide contacts. This nucleotide specificity further supports the significance of the UBXD1-bound open state. These data are now shown in Fig. 2c,d and Extended Data Fig. 4a-c, and discussed in lines: 232-247. Finally, we also note that our analysis of the ASPL adapter reveals that the UBX-lariat domain is conserved and serves related functions in these two adapters by destabilizing the p97 interprotomer interface, further supporting ring-opening as a conserved function. However, the biological role of UBXD1 is expected to be unique and related to p97 substrate processing given its asymmetric multi-domain interactions tether the open state and retain p97 in the hexamer form.

“Some interesting ideas are proposed in the Discussion about the potential role of the open state during substrate translocation or release, but there is no indication that the complex interacts with substrate.” The role of UBXD1 in p97 substrate translocation (if any) is indeed poorly understood. However, several lines of evidence are suggestive to us of a substrate-related function. First, UBXD1 uses its helical lariat to disrupt p97 interprotomer interactions but p97 complexes remain hexameric. In contrast, the ASPL adapter, which also contains a helical lariat, completely disassembles p97 hexamers, and therefore renders substrate processing by p97 impossible. As substrate translocation is the only known function of p97 and appears only to occur in hexamers, it appears likely that UBXD1 functions in this process. Additionally, during clearance of damaged lysosomes by macroautophagy (lysophagy), UBXD1 is observed to coordinate with additional p97 adapters, namely the deubiquitinase YOD1 and the ubiquitin-binding adapter PLAA⁴. Importantly, these adapters were observed to simultaneously form a complex with p97, indicating that p97:UBXD1 complexes in isolation or with these additional adapters interact with ubiquitylated substrates. Further evidence that p97:UBXD1 complexes interact with substrate comes from a proteomic study of UBXD1 interactions by the Deshaies group, which showed that UBXD1 interacts with ERGIC-53, an integral membrane protein involved in protein trafficking⁵. Importantly, this interaction was dependent on the presence of the H1/H2 region of UBXD1 (which is not resolved in our structures) and on p97 ATPase activity, but not on cellular ubiquitylation activity. These results suggest that UBXD1 might additionally act as a ubiquitin-independent substrate adapter. Finally, as discussed above, a recent study from the Meyer and Beuck groups (published during revisions of this manuscript) reported that the UBX domain of UBXD1 binds ubiquitin, indicating a direct substrate interaction¹. Thus, UBXD1 may act to recruit ubiquitylated substrates to p97 or interact with them post-translocation, potentially to facilitate interaction with a proteasomal shuttling factor.

To highlight the implications of these findings on UBXD1's role in substrate translocation, we have added text to the Discussion emphasizing these points, lines 524-542.

Given this supportive evidence, we are actively pursuing structural/functional characterizations of UBXD1 in the context of p97 substrate processing. However, given the substantial challenges of this system and likelihood that distinct, uncharacterized substrates and adapters associated with UBXD1-specific pathways (e.g. the lysosomal damage pathway) would be required for proper reconstitution, we feel this is beyond the scope of this manuscript. Indeed, though UBXD1 has been observed to simultaneously interact with an actively processing Ufd1/Npl4/p97 complex for which biochemical assays exist¹, UBXD1 and the Ufd1/Npl4 adapter are not observed to function together in at least one UBXD1 pathway⁴. Therefore, any conclusions from these experiments are likely uninformative. Thus, novel methods would need to be developed to reconstitute biologically relevant substrate- and UBXD1-bound complexes.

“The manuscript would be strengthened by addressing how the structure fits in the context of regulating p97 function in UBXD1-associated pathways. Previous studies indicate a relationship between UBXD1 and a variety of cellular functions, including turnover of damaged organelles, ERAD, endolysosomal sorting, and so forth. It would be fascinating to know the role of the UBXD1 in the functional cycle of p97 in such pathways.” We agree these are fascinating questions given what we now know about UBXD1's highly unusual mode of interaction. We thank the Reviewer for bringing up these important points. As discussed above, we have more strongly connected our structures with previous studies of UBXD1-mediated regulation of p97 activity in various cellular pathways (lines 524-542). While we agree that uncovering the exact role of UBXD1 binding in these pathways is of extreme interest, we consider such studies would require substantial proteomic and cellular assay development to define additional factors and functions related to UBXD1's interactions. We are currently pursuing experiments *in vitro* and in cells to further investigate the role of UBXD1 and other adapters in lysophagy, given the compelling observations of the Meyer group⁴ discussed above. However, given the major unresolved questions in this process, including the identity of the native substrate(s) and the operative p97:multi-adapter complex in UBXD1-involved pathways, these experiments are ongoing and incomplete.

Reviewer 2:

“One aspect of the study that is a little more challenging to understand is the observation of a VIM-only complex, and as mentioned below some more information could be given regarding this state”; p. 6, line 180: it's not clear how the authors conclude the additional helical density in the NTD cleft of all protomers is from UBXD1 VIM. Is the resolution for this helix sufficient that this density could be readily mapped as VIM? What dictates whether VIM displaces the UBX and other contacts? Mechanistically, how does it prevent UBX binding; does it have higher affinity and bind to an overlapping site for example?

The Reviewer brings up an important point about the identity of the helical density present in all NTDs under saturating UBXD1 conditions. We modeled this density as a VIM by orienting the

helix from the AlphaFold2 model in the density such that the conserved Arg residue present in nearly all VIMs (Arg62 in UBXD1) makes the same contact with the backbone carbonyl of Ala142 observed in a previous NTD-VIM structure⁶. Given the extreme conservation of this Arg residue (Fig. 1a for Reviewers), we reasoned that all VIMs are likely to feature this interaction. Using this approach, the UBXD1 VIM fits well into the density, thus supporting our claim that this helix corresponds to the VIM (Extended Data Fig. 5b). Furthermore, no other helix in UBXD1 is identified or predicted to bind the NTD cleft (Fig. 1b), though we acknowledge that the H1/H2 region was not observed in our structures (nor other regions predicted to be unstructured in the AlphaFold2 model). However, H1/H2 is observed by NMR to bind to the NTD/D1 interface of p97^{7,8}, and is thus unlikely to be responsible for the helical density in the NTD cleft. Additionally, mutation of other helical regions in UBXD1 (the helical lariat and H4) maintains the density in the NTD cleft, further supporting its assignment as a VIM (Extended Data Fig. 7m,n). However, to further support our claims we mutated four residues in the UBXD1 VIM that have been demonstrated to abolish binding to p97 (A58L, A59L, R62A, L63A)^{9,10} and determined cryo-EM structures using identical conditions as for the structures presented in Fig. 1, i.e. 2:1 UBXD1:p97 monomer (Fig. 1b for Reviewers). In a consensus map, no helical density is present in any p97 NTD, supporting that the density in structures using UBXD1 constructs with a wild-type VIM indeed corresponds to the VIM. Additional lowresolution density corresponding to other UBXD1 domains is present, indicating that the VIM interaction is specifically abolished with this mutation.

Why all p97 NTDs not bound by a UBX have VIM density (in saturating conditions) is an interesting question. As noted in the Discussion (lines 562-567), the affinities for p97 of individual UBXD1 domains are considerably weaker than in other p97 adapters, likely due to the multivalent nature of the p97-UBXD1 interaction that appears driven by avidity of multiple interacting domains. Indeed, (as discussed above) several previous studies have proposed that the UBX domain of UBXD1 is somewhat degenerate and does not bind the NTDs, identifying that the isolated UBX domain from UBXD1 does not interact with p97¹¹⁻¹³. Thus, our work is the first to reveal that UBX does in fact bind the NTD. Based on our structures, this interaction likely requires additional elements, such as the helical lariat, to further stabilize interactions through avidity effects. Supporting this, mutation of the lariat causes the complete disappearance of UBX density by cryo-EM (Fig. 7a,c). This additional binding requirement likely explains why UBX domains are not observed on NTDs from otherwise unbound p97 protomers: the p97 hexamer would need to be remodeled to accommodate lariats for each UBX domain. As observed in our cryo-EM data, a maximum of two UBX-lariat modules with appropriate spacing can bind per hexamer (Extended Data Fig. 1i), and only one in the open state (Fig. 1f), likely due to the hexamer rearrangements caused by the lariat (Fig. 2a,b). In contrast, binding of a VIM to an NTD does not induce any significant conformational changes in the hexamer and appears not to require other UBXD1 domains, thus explaining why many such domains can bind open NTDs with minimal effect.

Fig. 1 for Reviewers. Analysis of UBXD1 VIM sequence

(a) Alignment of all human VIM sequences. Canonical VIM residues ($RX_5A_2X_2R$) are highlighted in red, and the highly conserved Arg residue is indicated (*). (b) Sharpened cryo-EM maps of the P1 NTD from p97:UBXD1^{closed} (left) and of a representative NTD from a dataset of p97 and UBXD1 with four VIM mutations (right), showing loss of helical density.

“It is somewhat surprising that the LX/H4X mutations didn’t have a bigger effect in Figure 5e. Inclusion of experimental data that the LX and H4X mutations led to loss of binding for these regions might strengthen this part of the manuscript albeit the loss of density in the cryoEM structures does provide some experimental support for the efficacy of these mutations.”

We agree that the effects of mutating the lariat and/or H4 on inhibition of p97 ATPase activity are modest.

However, when considering the numerous other UBXD1 domains that contribute to binding and perhaps to ATPase inhibition, this is perhaps not unexpected. Indeed, analytical SEC reveals that the binding affinity of these mutants for p97 is not grossly impaired (Extended Data Fig. 7d,e), indicating that other domains (H1/H2, VIM, PUB) might play a role in the observed biochemical activity. Like the Reviewer, we consider loss of cryoEM density for the mutated regions to be supportive that binding by the lariat is indeed impaired or abolished.

Given the modest effects of the mutations pointed out by the Reviewer, we sought to characterize the effects of additional UBXD1 truncations on p97 ATPase activity, with the goal of identifying constructs with more dramatic loss of p97 ATPase inhibition that would provide more insight about the mechanism. To that end, we purified and tested two UBXD1 truncations individually and in combination, the results of which appear in a revised version of what is now Fig. 6 and associated text (lines 426-435). The first of these is a previously characterized construct containing the H1/H2, VIM, and H4 regions termed UBXD1-N^{7,8,10}, and the second, termed UBXD1-C, contains the PUB, lariat, and UBX domains. Based on SEC used during purification we conclude that these truncations are folded and stable. Both constructs have markedly impaired inhibitory activity, and do not saturate at the highest concentration tested (these and other UBXD1 constructs are not soluble in assay conditions at concentrations higher than ~5 μ M). This is in strong contrast to the results with individual domains mutated, and

indicates that domains in both the N- and C-terminal regions of UBXD1 are necessary for potent inhibition. Next, to determine whether ATPase inhibition depends on all UBXD1 domains being present in a single polypeptide, we incubated 1:1 mixtures of UBXD1-N and UBXD1-C with p97 and measured ATPase activity as before. This mixture had the same inhibition profile as UBXD1-C, indicating that all UBXD1 domains need to be tethered to potentially interfere with hydrolysis. Of note, the reduced affinity of UBXD1-N^{6,7,10} and likely of UBXD1-C compared to full-length UBXD1 might also contribute to the reduced potency. Regardless, it appears that tethering across multiple p97 protomers by a single UBXD1 molecule is necessary for full inhibitory activity.

Reviewer 2 additional points:

p. 2, line 60; 'subtracts' should be 'substrates'.

Agreed and corrected.

Extended Data Fig. 1b: molecular weight markers should be included on the lower image panels.

We have added molecular weight markers on the lower panels, as well as on those in Extended Data Fig. 7e.

p. 4, line 118-119: this Reviewer can't see the stated 'modest shift in elution' for ADP compared to

ATPgamaS. Either clarification is needed or perhaps this statement should be deleted. This also appears on p. 7, line 193-194.

Indeed, the shift of the peak corresponding to the complex when incubated with ATPyS relative to ADP is modest at best; we have removed this language so as not to over-interpret our data.

Fig. 1e,f: the two previously uncharacterized helices (in orange?) and N-terminal linker (not clear but probably also in orange?) should be labeled as it's not easy to visualize them in this figure.

We have added labels for the helical lariat in Fig. 1e,f on the side views of the sharpened maps. To further facilitate identification of these and other elements in UBXD1, we have added labels for the lariat and H4 on the AlphaFold2 model of UBXD1 in Fig. 1b.

p. 7, line 211: 'seem' should probably be 'seam'.

Agreed and corrected.

References

1. Blueggel, M. *et al.* The UBX domain in UBXD1 organizes ubiquitin binding at the C-terminus of the VCP/p97 AAA-ATPase. *Nat. Commun.* 14, 3258 (2023).
2. Conte, L. L., Chothia, C. & Janin, J. The atomic structure of protein-protein recognition sites Edited by A. R. Fersht. *J Mol Biol* 285, 2177–2198 (1999).

3. Meyer, H. & Boom, J. van den. Targeting of client proteins to the VCP/p97/Cdc48 unfolding machine. *Frontiers Mol Biosci* 10, 1142989 (2023).
4. Papadopoulos, C. *et al.* VCP/p97 cooperates with YOD1, UBXD1 and PLAA to drive clearance of ruptured lysosomes by autophagy. *Embo J* 36, 135–150 (2016).
5. Haines, D. S. *et al.* Protein Interaction Profiling of the p97 Adaptor UBXD1 Points to a Role for the Complex in Modulating ERGIC-53 Trafficking*. *Mol Cell Proteomics* 11, M111.016444 (2012).
6. Hänzelmann, P. & Schindelin, H. The Structural and Functional Basis of the p97/Valosin-containing Protein (VCP)-interacting Motif (VIM) MUTUALLY EXCLUSIVE BINDING OF COFACTORS TO THE N-TERMINAL DOMAIN OF p97*. *J Biol Chem* 286, 38679–38690 (2011).
7. Schuetz, A. K. & Kay, L. E. A Dynamic molecular basis for malfunction in disease mutants of p97/VCP. *Elife* 5, e20143 (2016).
8. Schütz, A. K., Rennella, E. & Kay, L. E. Exploiting conformational plasticity in the AAA+ protein VCP/p97 to modify function. *Proc National Acad Sci* 114, E6822–E6829 (2017).
9. Stapf, C., Cartwright, E., Bycroft, M., Hofmann, K. & Buchberger, A. The General Definition of the p97/Valosin-containing Protein (VCP)-interacting Motif (VIM) Delineates a New Family of p97 Cofactors*. *J Biol Chem* 286, 38670–38678 (2011).
10. Trusch, F. *et al.* The N-terminal Region of the Ubiquitin Regulatory X (UBX) Domain-containing Protein 1 (UBXD1) Modulates Interdomain Communication within the Valosin-containing Protein p97. *J Biol Chem* 290, 29414–27 (2015).
11. Bento, A. C. *et al.* UBXD1 is a mitochondrial recruitment factor for p97/VCP and promotes mitophagy. *Sci Rep-uk* 8, 12415 (2018).
12. Madsen, L. *et al.* Ubxd1 is a novel co-factor of the human p97 ATPase. *Int J Biochem Cell Biology* 40, 2927–2942 (2008).
13. Kern, M., Fernandez-Sáiz, V., Schäfer, Z. & Buchberger, A. UBXD1 binds p97 through two independent binding sites. *Biochem Biophys Res Co* 380, 303–307 (2009).

Decision Letter, first revision:**Message:** Our ref: NSMB-A47289A

13th Jul 2023

Dear Dr. Southworth,

Thank you for submitting your revised manuscript "The p97/VCP adapter UBXD1 drives AAA+ remodeling and ring opening through multi-domain tethered interactions" (NSMB-A47289A). It has now been seen by the original referees and their comments are below. The reviewers find that the paper has improved in revision, and therefore we'll be happy in principle to publish it in Nature Structural & Molecular Biology, pending minor revisions to satisfy the referees' final requests and to comply with our editorial and formatting guidelines.

To facilitate our work at this stage, it is important that we have a copy of the main text as a word file. If you could please send along a word version of this file as soon as possible, we would greatly appreciate it; please make sure to copy the NSMB account (cc'ed above).

Sincerely,
Kat

Katarzyna Ciazynska
(she/her)
Associate Editor
Nature Structural & Molecular Biology
<https://orcid.org/0000-0002-9899-2428>

Reviewer #1 (Remarks to the Author):

The revised manuscript by Braxton et al. has addressed my concerns. The study is excellent and will be well received by the protein homeostasis community. Specifically, the revised text brings a more satisfying connection with the biological significance gleaned from the structural work, as does the new dataset processed of substoichiometric UBXD1:p97 complexes. I support publication.

Reviewer #2 (Remarks to the Author):

All my concerns have been addressed and I remain highly enthusiastic towards the publication of this manuscript.

Final Decision Letter:**Message** 14th Sep 2023

:

Dear Dr. Southworth,

We are now happy to accept your revised paper "The p97/VCP adapter UBXD1 drives AAA+ remodeling and ring opening through multi-domain tethered interactions" for publication as an Article in Nature Structural & Molecular Biology.

As soon as your article is published, you can generate your shareable link by entering the DOI of your article here: http://authors.springernature.com/share. Corresponding authors will also receive an automated email with the shareable link

Your paper will be published online soon after we receive proof corrections and will appear

in print in the next available issue. You can find out your date of online publication by contacting the production team shortly after sending your proof corrections. Content is published online weekly on Mondays and Thursdays, and the embargo is set at 16:00 London time (GMT)/11:00 am US Eastern time (EST) on the day of publication. Now is the time to inform your Public Relations or Press Office about your paper, as they might be interested in promoting its publication. This will allow them time to prepare an accurate and satisfactory press release. Include your manuscript tracking number (NSMB-A47289B) and our journal name, which they will need when they contact our press office.

About one week before your paper is published online, we shall be distributing a press release to news organizations worldwide, which may very well include details of your work. We are happy for your institution or funding agency to prepare its own press release, but it must mention the embargo date and Nature Structural & Molecular Biology. If you or your Press Office have any enquiries in the meantime, please contact press@nature.com.

Please note that *Nature Structural & Molecular Biology* is a Transformative Journal (TJ). Authors may publish their research with us through the traditional subscription access route or make their paper immediately open access through payment of an article-processing charge (APC). Authors will not be required to make a final decision about access to their article until it has been accepted. [Find out more about Transformative Journals](https://www.springernature.com/gp/open-research/transformative-journals)

Authors may need to take specific actions to achieve [compliance](https://www.springernature.com/gp/open-research/funding/policy-compliance-faqs) with funder and institutional open access mandates. If your research is supported by a funder that requires immediate open access (e.g. according to [Plan S principles](https://www.springernature.com/gp/open-research/plan-s-compliance)) then you should select the gold OA route, and we will

direct you to the compliant route where possible. For authors selecting the subscription publication route, the journal's standard licensing terms will need to be accepted, including [self-archiving policies](https://www.springernature.com/gp/open-research/policies/journal-policies). Those licensing terms will supersede any other terms that the author or any third party may assert apply to any version of the manuscript.

Sincerely,
Kat

Katarzyna Ciazynska
(she/her)
Associate Editor
Nature Structural & Molecular Biology
<https://orcid.org/0000-0002-9899-2428>
